# Structures of a deltacoronavirus spike protein bound to porcine and human receptors

Weiwei Ji[1,7], Qi Peng[2,3,4,7], Xueqiong Fang[1], Zehou Li[1], Yaxin Li[1], Cunfa Xu[5], Shuqing Zhao[2,3,4], Jizong Li[2,3,4], Rong Chen[2], Guoxiang Mo[1], Zhanyong Wei[6], Ying Xu[1✉], Bin Li[2,3,4✉] & Shuijun Zhang[1✉]

Porcine deltacoronavirus (PDCoV) can experimentally infect a variety of animals. Human infection by PDCoV has also been reported. Consistently, PDCoV can use aminopeptidase N (APN) from different host species as receptors to enter cells. To understand this broad receptor usage and interspecies transmission of PDCoV, we determined the crystal structures of the receptor binding domain (RBD) of PDCoV spike protein bound to human APN (hAPN) and porcine APN (pAPN), respectively. The structures of the two complexes exhibit high similarity. PDCoV RBD binds to common regions on hAPN and pAPN, which are different from the sites engaged by two alphacoronaviruses: HCoV-229E and porcine respiratory coronavirus (PRCoV). Based on structure guided mutagenesis, we identified conserved residues on hAPN and pAPN that are essential for PDCoV binding and infection. We report the detailed mechanism for how a deltacoronavirus recognizes homologous receptors and provide insights into the cross-species transmission of PDCoV.

[1] College of Life Sciences, Nanjing Agricultural University, Nanjing 210095, China. [2] Institute of Veterinary Medicine, Jiangsu Academy of Agricultural Sciences, Key Laboratory of Veterinary Biological Engineering and Technology, Ministry of Agriculture, Nanjing 210014, China. [3] Jiangsu Key Laboratory for Food Quality and Safety-State Key Laboratory Cultivation Base of Ministry of Science and Technology, Nanjing 210014, China. [4] Jiangsu Coinnovation Center for Prevention and Control of Important Animal Infectious Diseases and Zoonoses, Yangzhou University, Yangzhou 225000, China. [5] Central Laboratory of Jiangsu Academy of Agricultural Sciences, Nanjing 210014, China. [6] College of Veterinary Medicine, Henan Agricultural University, Zhengzhou 450046, China. [7] These authors contributed equally: Weiwei Ji, Qi Peng. ✉email: bioyingxu@njau.edu.cn; libinana@126.com; zhangsj@njau.edu.cn

Coronaviruses (CoVs) are enveloped RNA viruses categorized into four genera, *Alphacoronavirus*, *Betacoronavirus*, *Gammacoronavirus*, and *Deltacoronavirus*[1]. They cause mild to severe respiratory and gastrointestinal diseases in humans and animals[1]. The betacoronaviruses, severe acute respiratory syndrome CoV (SARS-CoV), Middle East respiratory syndrome CoV (MERS-CoV), and SARS-CoV-2 have caused large-scale pandemics in 2003, 2012 and 2019, respectively, resulting in significant morbidity and mortality in the human population[2]. The ongoing SARS-CoV-2 pandemic has now claimed more than 5 million lives worldwide[3]. Porcine deltacoronavirus (PDCoV), a member of *Deltacoronavirus* genus, causes severe diarrhea and vomiting in piglets[4]. PDCoV was first identified in Hong Kong, China, in 2012[5], and outbreak of PDCoV has since then been reported in many countries[6–10]. PDCoV, along with two alpha-coronaviruses, porcine epidemic diarrhea virus (PEDV) and transmissible gastroenteritis virus (TGEV), have caused a high number of deaths among piglets and therefore pose serious threats to the pork industry[4]. Recently, human infection of PDCoV was also reported and the virus was isolated from the plasma sample of children with acute febrile illness[11]. This highlights the risk of PDCoV transmission among the human population.

The initial step in coronavirus infection is the binding of the viral spike protein (S protein) to the receptor on the host cell surface[12]. The S protein is a type I transmembrane protein and its ectodomain consists of two subunits, S1 and S2. The receptor-binding domain (RBD) is located in S1 subunit[12]. The binding of the S1 subunit to viral receptors leads to a large conformational change in the S2 subunit that drives the fusion of viral and cellular membranes[13]. Coronaviruses have evolved to adopt a complex pattern of receptor recognition. Though both belong to *Betacoronavirus* genus, SARS-CoV and MERS-CoV engage angiotensin-converting enzyme 2 (ACE2) and dipeptidyl peptidase 4 (DDP4) as receptors, respectively[14,15]. Unexpectedly, human coronavirus NL63 (HCoV-NL63), an alphacoronavirus, recognizes the same receptor (ACE2) as betacoronavirus SARS-CoV[16]. Similarly, APN is a common receptor for alphacoronaviruses HCoV-229E and TGEV[17,18]. Recently, aminopeptidase N (APN) has also been identified as an entry receptor for deltacoronavirus PDCoV by two research groups[19,20]. Structural studies indicate that HCoV-229E and TGEV bind to different regions on APN[21–23], but how PDCoV interacts with APN is not yet clear.

Coronaviruses sporadically break the species barrier and spill over to new hosts[1]. The interaction between the viral S protein and the receptor largely determines the host spectrum and tissue tropism of coronaviruses, thus providing pivotal roles in the cross-species transmission of virus[1,12]. Coronaviruses, including SARS-CoV, MERS-CoV and SARS-CoV-2, could engage homologous receptors derived from multiple species[24–26]. Broad receptor engagement provides essential conditions for coronaviruses to jump across species, which have been extensively studied in SARS-CoV[27–29]. Mutation of only a few residues on the spike protein could alter the receptor usage of coronavirus, resulting in host range expansion. A single K479N mutation on SARS-CoV S protein enhanced virus affinity to human ACE2 receptor, which facilitated the transmission of SARS-CoV from palm civets to humans[14,28]. Similar to SARS-CoV, PDCoV also exhibits broad receptor usage and could employ porcine, human, and avian APN to enter cells[19]. Moreover, PDCoV could experimentally infect chickens, calves, and mice[30–33]. Reminiscent of symptoms observed in infected pigs, PDCoV infection in chickens also causes diarrhea[30,31]. Phylogenetic analysis reveals that PDCoV is closely related to sparrow CoV HKU17, bulbul CoV HKU11 and munia CoV HKU13[5]. These findings support the hypothesis that PDCoV may have evolved from an avian deltacoronavirus ancestor[5]. A

recent report of PDCoV infection among children has sounded alarmed about the further adaptation of the virus to humans[11]. The virus-receptor interaction of alpha- and betacoronaviruses has been studied extensively[12,22,34–36]. However, it is not yet clear how deltacoronaviruses recognize their receptors.

Here, we report the crystal structures of PDCoV RBD complexed with human and porcine APN receptors, respectively. The structures show that PDCoV binds to conserved regions on human and porcine APN, which explains the broad receptor usage of the virus. The PDCoV binding site on APN differs from that targeted by either TGEV or HCoV-229E. Therefore, in comparison with alphacoronavirus, deltacoronavirus have evolved to acquire distinct receptor interaction mode. Based on structure-guided mutagenesis, we also identified conserved residues on homologous APN molecules that affect PDCoV infection of cells. Unexpectedly, PDCoV RBD shows a higher binding affinity to human APN than to porcine APN. Taken together, these results provide the structural basis for receptor recognition by PDCoV, highlighting the cross-species potential of PDCoV and the risk of virus adaptation to the human population.

## Results

**Structures of PDCoV RBD complexed with human and porcine APN.** The PDCoV RBD is located at the C terminal of the S1 subunit of S protein (Fig. 1a). Previous research revealed that PDCoV RBD bound to cells overexpressing pAPN[19,20]. In addition to pAPN, PDCoV could also use human and avian APN as receptors to enter cells[19]. Therefore, in this study, to characterize the interaction between PDCoV and receptor APN, we solved the crystal structures of PDCoV RBD complexed with human and porcine APN, respectively. We expressed PDCoV RBD (residue 300–419), the ectodomains of human APN (hAPN, residue 62–963), and pAPN (residue 58–961) in Hi5 insect cells, purified them by Ni-NTA affinity purification and gel filtration. Crystals of PDCoV RBD bound to hAPN or pAPN were obtained by co-crystallization of respective proteins. The structures of PDCoV RBD-hAPN and PDCoV RBD-pAPN complexes were determined to be 3.1 and 2.7 Å, respectively (Table 1). PDCoV RBD–pAPN complex contains an APN dimer in the asymmetric unit (ASU) but only one monomer is associated with PDCoV RBD, while PDCoV RBD–hAPN complex has three RBD–hAPN heterodimers in the ASU. The structures of the two complexes exhibit high similarity, with a root mean square deviation (RMSD) of 0.63 Å over 976 equivalent Cα atoms (Fig. 1b–d). The shape complementarity values, calculated with Sc[37], are 0.66 and 0.56 for PDCoV RBD–hAPN and PDCoV RBD–pAPN, suggesting PDCoV shows better complementarity for hAPN in terms of shape. The PDCoV RBD adopts β-barrel structure, similar to the RBD of alphacoronaviruses PRCoV, a TGEV variant, and HCoV-229E, both of which also use APN as a cellular receptor (Supplementary Fig. 1a–c). The receptor-binding motifs (RBMs) in the PDCoV RBD consist of 4 β-strands (β2, β4–β6) and 2 connecting loops (β1–β2 loop and β5–β6 loop) (Fig. 1b, c). Unlike the RBMs of PRCoV and HCoV-229E, the receptor-binding loops of PDCoV RBD are much shorter (Supplementary Fig. 1a–c). APN is a zinc-dependent metalloenzyme that hydrolyzes the N-terminus of bioactive peptides, playing important roles in angiogenesis, regulation of blood pressure and other biologic processes[38]. Residues on APN that contact PDCoV RBMs are located on the outer surface and are distant from the active site of the enzyme (Supplementary Fig. 2a). pAPN exhibits open conformation in the previously reported structure of PRCoV RBD–pAPN complex[21], in which domain IV moves away from Domain I and Domain II, thus creating a tunnel leading to the catalytic site (Supplementary Fig. 2b, c). However, in this study,

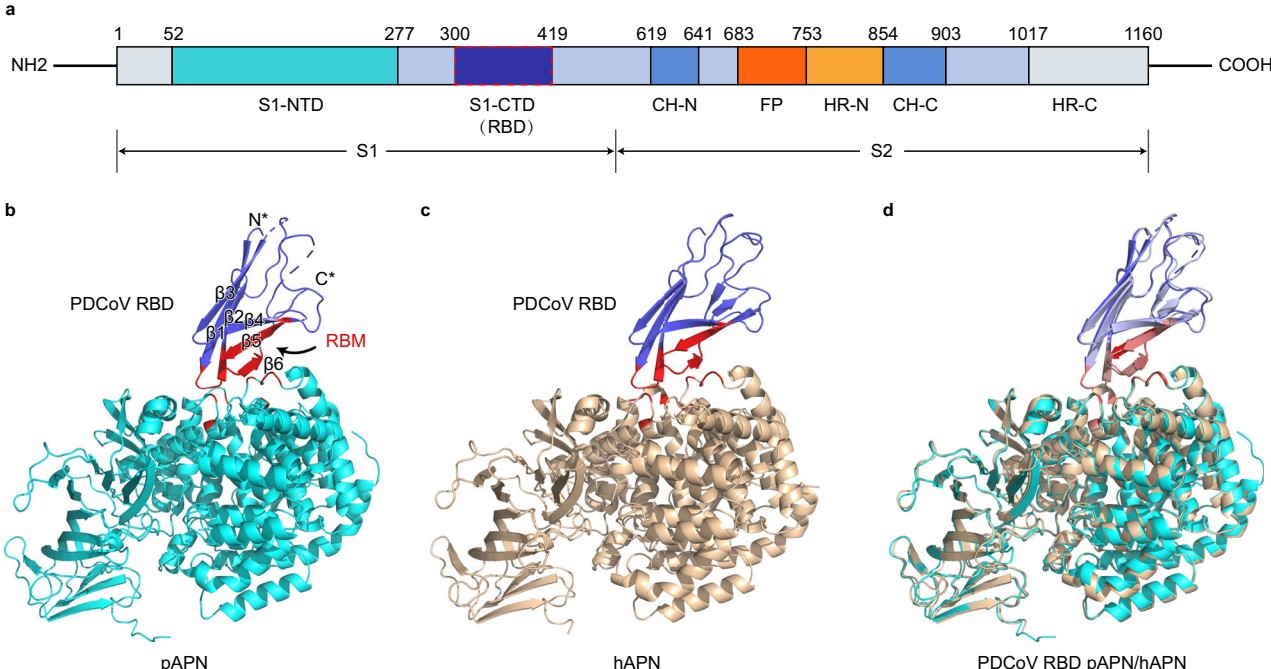

**Fig. 1 Overall structure of PDCoV RBD in complex with human APN (hAPN) or porcine APN (pAPN). a** Schematic diagram of the PDCoV spike (S) protein ectodomain. S1-NTD, N-terminal domain of S1. S1-CTD, C-terminal domain of S1. RBD receptor binding domain. CH-N N-terminal central helix. CH-C C-terminal central helix, FP fusion peptide, HR-N N-terminal heptad repeat, HR-C C-terminal heptad repeat. **b** Crystal structure of PDCoV RBD bound to pAPN. PDCoV RBD and pAPN are colored in purple and cyan, respectively. The receptor-binding motifs (RBMs) on PDCoV RBD and virus binding motifs (VBMs) on pAPN are shown in red. PDCoV RBD adopts β barrel structure and contains six β-strands. **c** Crystal structure of PDCoV RBD bound to hAPN. PDCoV RBD and hAPN are colored in purple and brown, respectively. RBMs and VBMs are represented as in (**b**). **d** Superposition of PDCoV RBD–pAPN and PDCoV RBD–hAPN complexes. The two structures exhibit high similarity and the root mean square deviation (RMSD) between them is 0.63 Å.

both pAPN and hAPN crystallized in the complexes adopt close conformations, in which DII and DIV are closer to each other, thus limiting access to the active site of the enzyme (Supplementary Fig. 2a, c).

**PDCoV and alphacoronaviruses bind to different sites on APN.** PDCoV shares the same receptor, APN, with two alphacoronaviruses, PRCoV and HCoV-229E[19,20]. However, PDCoV binds to regions on APN that differ from the sites bound by either PRCoV (Fig. 2a) or HCoV-229E (Fig. 2b), evidence of divergent receptor interaction of deltacoronaviruses. PDCoV-hAPN interaction results in a large buried surface area (BSA) of 1816 Å$^2$ (928 Å$^2$ on the PDCoV RBD and 888 Å$^2$ on hAPN). Approximately the same BSA (1811 Å$^2$) has been observed at PDCoV RBD–pAPN interface (921 Å$^2$ on the PDCoV RBD and 890 Å$^2$ on pAPN). PDCoV binds to similar regions on hAPN and pAPN, spanning domain II (DII) and domain IV (DIV) of APN (Fig. 2a, b). These two domains contribute to 38% and 62% of the total BSA on hAPN, respectively. However, the DII of pAPN contributes very little (6% of the total BSA) to the footprint of PRCoV. Instead, PRCoV almost entirely docks on DIV (Fig. 2a), which accounts for 94% of total BSA on pAPN. In contrast to PRCoV, DIV of APN is not involved in binding to HCoV-229E and the virus almost exclusively targets DII (Fig. 2b), with DII contributing to 95% of the total BSA on hAPN. The surface area buried between PDCoV RBD and hAPN is larger than that at the PRCoV RBD–pAPN (1461 Å$^2$) or HCoV-229E RBD–hAPN interface (994 Å$^2$). Within a distance cutoff of 4.5 Å, a total of 18 residues on PDCoV RBD are in contact with 25 residues on hAPN or 27 residues on pAPN (Fig. 2c, Supplementary Table 1). Only 1 out of the 25 residues on hAPN, D315, is engaged by both

PDCoV and HCoV-229E (Fig. 2c). As the footprints of PDCoV and PRCoV partially overlap on DIV of pAPN, 10 out of 27 residues are bound by both viruses (Fig. 2c). In addition, PDCoV RBD binding modes of pAPN and hAPN are highly similar (Fig. 2a, b). In summary, PDCoV binds to conserved regions on hAPN and pAPN that are different from the ones occupied by either HCoV-229E or PRCoV, which indicates that though sharing the same receptor with alphacoronaviruses, deltacoronavirus has evolved to acquire separate receptor binding mode.

**PDCoV recognizes conserved residues on hAPN and pAPN.** There are two PDCoV RBD interacting regions on APN (Fig. 2a, b). The first region involves the contact between DII of APN (α2 helix, α5 helix, and α6–α7 loop) and β1–β2 hairpin of PDCoV RBD (Fig. 3a). The ε-amino group of K379 on APN forms a network of hydrogen bond/salt bridge interactions with D317, F318, and E320 on PDCoV RBD (Fig. 3a). Our surface plasmon resonance (SPR) results show that K379A mutation on hAPN results in a more than 100-fold decrease of affinity (from $10^{-6}$ M to $10^{-4}$ M) to PDCoV RBD (Fig. 3b, Fig. 4a, b, Supplementary Table 2), while corresponding K374A mutation in pAPN abolishes its binding to PDCoV RBD (Fig. 3b, Fig. 4c, d, Supplementary Table 2). E426 and W429 on APN form a salt bridge and hydrogen bonding with R322 and E320 on PDCoV RBD, respectively (Fig. 3a). Accordingly, E426A and W429A mutations on hAPN, or respective E421A and W424A mutations on pAPN, cause 3-100 fold reduction of affinity (Fig. 3b, Fig. 4e–h, Supplementary Table 2). Additional interaction in this region involves the packing of Y316 on APN against F318 on PDCoV RBD (Fig. 3a) and disruption of this interaction also lead to a 10-fold decrease in affinity (Fig. 3b, Fig. 4i, Supplementary Table 2).

**Table 1 X-ray crystallographic data collection and refinement statistics.**

| | PDCoV RBD–pAPN complex | PDCoV RBD–hAPN complex |
|---|---|---|
| *Data collection* | | |
| Space group | P4₁2₁2 | C222 |
| *Cell dimensions* | | |
| $a$, $b$, $c$ (Å) | 186.64, 186.64, 173.82 | 201.23, 347.69, 255.11 |
| $\alpha$, $\beta$, $\gamma$ (°) | 90.00, 90.00, 90.00 | 90.00, 90.00, 90.00 |
| Resolution (Å)[a] | 50.00-2.69 | 50.00-3.10 |
| | (2.85-2.69) | (3.29-3.10) |
| Unique reflections | 84,784 (12,941) | 160,248 (25,301) |
| $R_{merge}$[a,b] | 0.193 (2.344) | 0.358 (1.183) |
| $R_{pim}$ | 0.062 (0.855) | 0.156 (0.564) |
| CC1/2 | 0.997 (0.472) | 0.964 (0.583) |
| $I/\sigma(I)$[a] | 10.69 (1.12) | 5.28 (1.45) |
| Completeness (%)[a] | 99.2 (95.4) | 99.4 (98.0) |
| Redundancy[a] | 10.68 (10.59) | 6.1 (6.2) |
| *Refinement* | | |
| Resolution (Å) | 42.65-2.69 | 49.36-3.10 |
| No. of reflections | 84,429 | 160,182 |
| $R_{work}/R_{free}$[c] | 0.198/0.230 | 0.259/0.278 |
| *No. of atoms* | | |
| Protein | 15,198 | 24,096 |
| Ligand/Ion | 382 | 441 |
| Water | 35 | 0 |
| *B-factors (Å²)* | | |
| Protein | 64.99 | 36.60 |
| Ligand/Ion | 90.13 | 63.11 |
| Water | 55.56 | 0 |
| *R.m.s. deviations* | | |
| Bond lengths (Å) | 0.002 | 0.003 |
| Bond angles (°) | 0.513 | 0.55 |
| *Ramachandran plot (%)[d]* | | |
| Favored | 95.47 | 94.41 |
| Allowed | 4.42 | 4.95 |
| Outliers | 0.11 | 0.64 |

[a]Values for the outmost resolution shell are given in parentheses.
[b]$R_{merge} = \Sigma_i\Sigma_{hkl}|I_i\langle I\rangle|/\Sigma_i\Sigma_{hkl}I_i$, where $I_i$ is the observed intensity and $\langle I\rangle$ is the average intensity from multiple measurements.
[c]$R_{work} = \Sigma||F_o||F_c||/\Sigma|F_o|$, where $F_o$ and $F_c$ are the structure-factor amplitudes from the data and the model, respectively. $R_{free}$ is the R factor for a subset (5%) of reflections that was selected prior to refinement calculations and was not included in the refinement.
[d]Ramachandran plots were generated by using the program MolProbity.

The second PDCoV RBD APN binding region mainly consists of the interactions between β5 and β6 hairpin of PDCoV RBD and DIV of APN (α19–α20 loop and α21–α22 loop). Residues on α19–α20 loop (R741–E742–I743–P744–E745) of APN make extensive contacts with residues on β5–β6 loop, β6 strand of PDCoV RBD (N397–Y398–L399–L400–R401), via hydrogen bonding and salt bridge (Fig. 3c, e). Most of the hydrogen bonds involve the main chain of residues on APN or RBD and therefore would be less dependent on the specific sequence of residues. Therefore, for residues that participate in hydrogen bonding, we only mutated those whose side chains were hydrogen donors or acceptors. E742A mutation on APN breaks the E742–R357 salt bridge (Fig. 3c) and leads to a 4-fold decrease in affinity (Fig. 3b, Fig. 4j, Supplementary Table 2). H789 on α21–α22 loop of APN is hydrogen-bonded with Y398 on PDCoV RBD as well (Fig. 3c). H789A and another mutation E745A only cause a slight decrease in affinity (Fig. 3b, Fig. 4k, l, Supplementary Table 2). Additionally, W396 on β5–β6 loop contacts L372 on DII of APN via hydrogen bonding (Fig. 3a). Interestingly, PDCoV RBD shows ~10-fold lower affinity to pAPN than to hAPN (Fig. 3b, d, Fig. 4a, c, Supplementary Table 2). Reciprocally, mutation of PDCoV RBD residues F318, E320, R322, R357, and Y398 also lead to 5- to

150-fold decrease in affinity to pAPN or hAPN (Fig. 3d, Supplementary Fig. 3a–h, Supplementary Fig. 4a–h, Supplementary Table 3). W396 alone accounts for 15% of the total buried surface area on PDCoV RBD and mutation of this residue abolishes the binding of PDCoV RBD to either hAPN or pAPN (Fig. 3d). Collectively, by using structure-guided mutagenesis, we have identified key residues that mediate the binding between PDCoV RBD and APN.

hAPN and pAPN share 79% amino acid sequence identity. Consistent with the similar PDCoV binding modes of hAPN and pAPN (Fig. 2a, b), most of the residues (19/25) contacting PDCoV RBD are strictly conserved between hAPN and pAPN (Fig. 2c). Moreover, key residues identified by the above mutagenesis studies are also conserved in positions on the structures of hAPN and pAPN (Fig. 3f). This is in accordance with previous studies indicating that PDCoV is able to use both hAPN and pAPN as entry receptors[19,20].

**Key residues on APN are required for PDCoV infection.** Based on the above structural and mutagenesis studies, we further characterize which residues on APN affect virus infection. We incubated PDCoV with wild-type or mutant APN proteins before infecting LLC-PK1 cells. Firstly, we show that both wild-type hAPN and pAPN inhibit virus infection in a dose-dependent manner at the early stage of virus infection (Supplementary Fig. 5a–d). Specifically, APN inhibits PDCoV adsorption to cells (Supplementary Fig. 5e, f). Secondly, to identify which residues on APN affect PDCoV infection, we incubated PDCoV with wildtype or mutant hAPN proteins at a concentration of 80 μg/ml at 37 °C for 1 h. Then LLC-PK1 cells were infected with APN-treated or mock-treated virus. The samples were collected for RT-qPCR and IFA. Our results show that compared to wildtype hAPN, Y316A, K379A, E426A, and W429A mutants block virus infection less efficiently (Fig. 5a–j). The extent to which these mutants inhibit virus infection decrease by ~2-fold compared to wild-type APN (Fig. 5j, k). Overexpression of hAPN mutants in BHK-21 cells reduced PDCoV infection in varying degrees compared to overexpression of hAPN wild-type protein (Fig. 6a–k). Consistent with the result from protein blocking assay, Y316A, K379A, E426A, or W429A led to ~70% reduction of infection (Fig. 6l, m), indicating these residues play important roles in virus receptor binding and infection. As these four residues are also completely conserved in pAPN and avian APN, we speculate that PDCoV jumps across species via binding to these key residues on homologous receptors from different hosts.

**Discussion**
Though the receptor recognition mechanisms of alpha- and betacoronaviruses have been studied extensively[12,22,23,34–36], the mechanism of how deltacoronaviruses bind to their receptors is not yet known. In this study, we first report the crystal structures of PDCoV RBD bound to human and porcine APN receptors (Fig. 1b, c). PDCoV footprints on hAPN or pAPN are different from those of HCoV-229E and PRCoV, providing evidence of novel receptor interaction of deltacoronaviruses (Fig. 2a, b). Based on structure-guided mutagenesis, we have identified conserved residues on porcine and human APN that are responsible for binding to PDCoV RBD (Fig. 3a–f). We have further carried out a PDCoV infection assay and shows that hAPN protein with structurally conserved residues mutated inhibits PDCoV much less efficiently compared to wild-type hAPN (Fig. 5a–k). When overexpressed on BHK-21 cells, APN with key residues mutated significantly inhibits PDCoV infection (Fig. 6a–k). In summary, via a combination of structure, biochemical, and virus infection

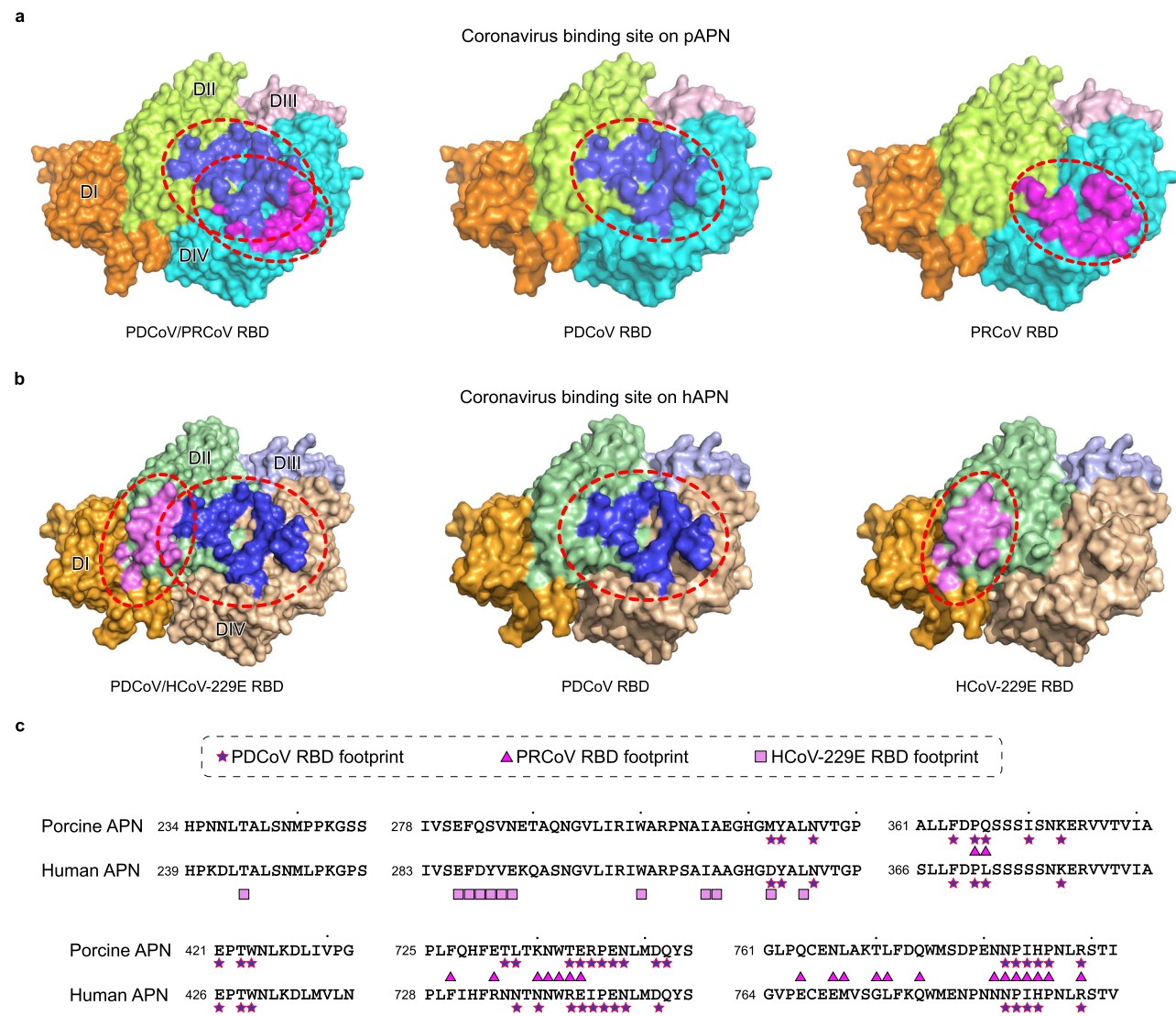

**Fig. 2 PDCoV binds to sites on APN that are different from alphacoronaviruses HCoV-229E and PRCoV. a** (Left panel) Overlay of PDCoV and PRCoV footprints on pAPN. pAPN is shown as surface representation, with DI–DIV colored in orange, light green, pink, and cyan respectively. Residues on pAPN contacting PDCoV RBD and PRCoV RBD are colored in blue and magenta, respectively. The boundaries of PDCoV and PRCoV footprints are circled with red dotted lines. (Middle panel) PDCoV RBD footprint on pAPN. (Right panel) PRCoV footprint on pAPN. PDCoV RBD footprint engages both DII and DIV of pAPN while PRCoV RBD largely targets DIV of pAPN. **b** (Left panel) Overlay of PDCoV and HCoV-229E footprints on hAPN. hAPN is depicted as surface representation. DI–DIV of hAPN are colored in orange, dark green, light blue, and brown, respectively. Residues on hAPN that bind to PDCoV RBD and HCoV-229E RBD are colored as in (**a**). The boundaries of PDCoV RBD and HCoV-229E RBD binding sites are circled with red dotted lines. (Middle panel) PDCoV RBD footprint on hAPN. (Right panel) HCoV-229E footprint on hAPN. PDCoV RBD binding sites on hAPN are also located both on DII and DIV of hAPN, whereas HCoV-229E RBD mainly binds to DII of APN. **c** Sequence alignment of pAPN and hAPN. Residues on pAPN/hAPN binding to PDCoV RBD, PRCoV RBD, and HCoV-229E RBD are marked according to the code of the key above the sequences.

assays, we have deciphered residues on APN receptors that play pivotal roles in PDCoV binding and infection.

Although the BSA at the interface of PDCoV RBD–hAPN or PDCoV RBD–pAPN is larger than that of HCoV-229E RBD–hAPN, the binding affinity of PDCoV RBD to hAPN or pAPN is lower compared to that of HCoV-229E RBD to hAPN. Electrostatic interactions were found experimentally and computationally to be important for protein binding[39]. Therefore, we compared the electrostatic potential, calculated using APBS[40] and PDB2PQR[41] packages, of PDCoV RBD–hAPN, PDCoV RBD–pAPN, and HCoV-229E RBD–hAPN (Supplementary Fig. 1d, e). There is only partial charge complementarity between PDCoV RBD and hAPN/pAPN. However, the RBMs on HCoV-229E RBD are fully positively charged and docks onto the negatively charged counterpart on hAPN. The stronger binding between hAPN and HCoV-229E RBD may thus result from complete charge complementarity between them. Since the electric potential distribution in pAPN and hAPN is highly similar, this unlikely accounts for the 10-fold difference in their affinity to PDCoV RBD. However, PDCoV RBD shows better shape complementarity for hAPN than pAPN (Sc value 0.66 vs 0.56), which may explain the stronger interaction between PDCoV RBD and hAPN. In addition, the sequence identity between pAPN and hAPN is 79%, and therefore other residues at the non-interacting interface may also lead to the difference in affinity.

The binding modes of PDCoV/pAPN and PDCoV/hAPN are very close. Moreover, PDCoV recognizes conserved residues on hAPN and pAPN. PDCoV strains isolated from children in Haiti were highly similar to the pig strains detected in China and America[11]. Specifically, the sequence identity of RBDs of different

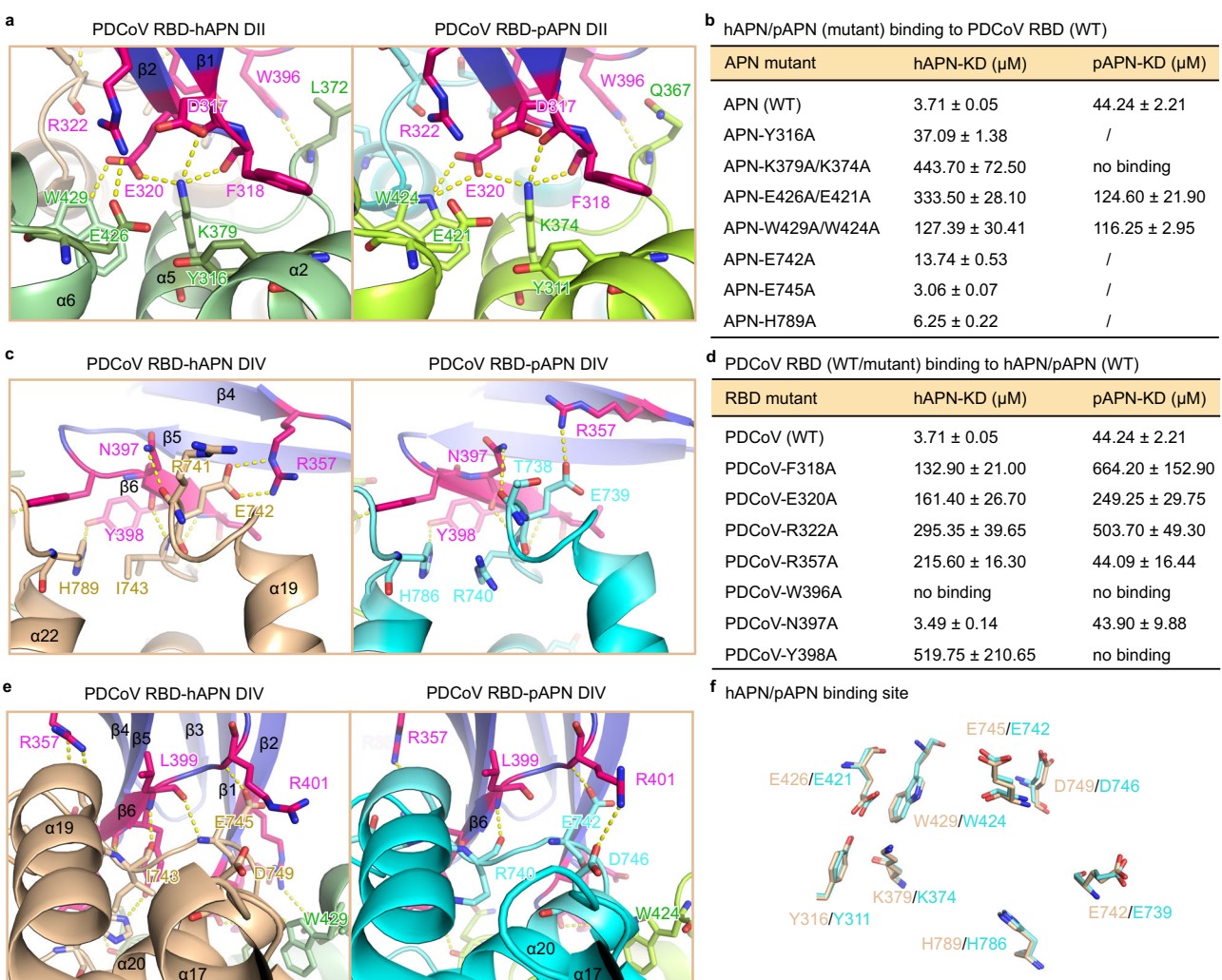

**Fig. 3 PDCoV recognizes conserved residues on human and porcine APN receptors. a, c, e** Atomic details of the interaction between PDCoV RBD and hAPN/pAPN. Contacting residues on respective proteins are represented as sticks, with nitrogen and oxygen atoms colored in blue and red, respectively. PDCoV RBD are shown in purple while different domains of hAPN/pAPN are colored as in Fig. 2a, b. **b** Analysis of the binding of PDCoV RBD to wildtype (WT) and mutant hAPN/pAPN proteins using surface plasmon resonance (SPR). The affinities of the interacting proteins are calculated as average values from two experiments, with standard deviation included. n.b. stands for no binding. / indicates the interactions have not been measured. **d** Analysis of the interaction between wildtype hAPN/pAPN and PDCoV RBD wildtype (WT)/mutant proteins using SPR. **f** Superposition of key residues on hAPN and pAPN that bind to PDCoV RBD. The side chains of these residues are well aligned in position. Values for binding affinity are provided as a Source Data file.

The following tables appear in the figure:

**b hAPN/pAPN (mutant) binding to PDCoV RBD (WT)**

| APN mutant | hAPN-KD (µM) | pAPN-KD (µM) |
|---|---|---|
| APN (WT) | 3.71 ± 0.05 | 44.24 ± 2.21 |
| APN-Y316A | 37.09 ± 1.38 | / |
| APN-K379A/K374A | 443.70 ± 72.50 | no binding |
| APN-E426A/E421A | 333.50 ± 28.10 | 124.60 ± 21.90 |
| APN-W429A/W424A | 127.39 ± 30.41 | 116.25 ± 2.95 |
| APN-E742A | 13.74 ± 0.53 | / |
| APN-E745A | 3.06 ± 0.07 | / |
| APN-H789A | 6.25 ± 0.22 | / |

**d PDCoV RBD (WT/mutant) binding to hAPN/pAPN (WT)**

| RBD mutant | hAPN-KD (µM) | pAPN-KD (µM) |
|---|---|---|
| PDCoV (WT) | 3.71 ± 0.05 | 44.24 ± 2.21 |
| PDCoV-F318A | 132.90 ± 21.00 | 664.20 ± 152.90 |
| PDCoV-E320A | 161.40 ± 26.70 | 249.25 ± 29.75 |
| PDCoV-R322A | 295.35 ± 39.65 | 503.70 ± 49.30 |
| PDCoV-R357A | 215.60 ± 16.30 | 44.09 ± 16.44 |
| PDCoV-W396A | no binding | no binding |
| PDCoV-N397A | 3.49 ± 0.14 | 43.90 ± 9.88 |
| PDCoV-Y398A | 519.75 ± 210.65 | no binding |

PDCoV strains, including the human infecting ones from Haiti, ranges from 96.1 to 100%. The RBM residues of these PDCoV strains are strictly conserved (Supplementary Fig. 6a). This indicates the risk of cross-species transmission of PDCoV between pig and human population[11]. Lednicky et al. found two mutations outside RBD of S1 subunit of the human infecting PDCoV strain[11], suggesting other regions on S protein may also affect the host range of PDCoV as well. Unlike PDCoV, TGEV cannot enter cells via hAPN while HCoV-229E does not bind to pAPN[42]. Docking of TGEV RBD on hAPN or HCoV-229E RBD on pAPN lead to steric clashes between the corresponding proteins[22]. Therefore, the interspecies transmission of HCoV-229E or TGEV between the human and pig populations is highly unlikely.

Phylogenetic studies suggest that PDCoV evolves from avian deltacoronaviruses[5]. To explore the possibility of avian deltacoronaviruses to use APN as a receptor, we compared PDCoV RBD and corresponding sequences from sparrow CoV HKU17, bulbul CoV HKU11 and munia CoV HKU13, all of which are closely related to PDCoV (Supplementary Fig. 6b). The sequence alignment shows that the RBM residues on PDCoV RBD show higher similarity to HKU11 and HKU13 than to HKU17 (Supplementary Fig. 6b). Meanwhile, a comparison of virus binding motifs (VBM) on APN reveals that most of these residues are largely identical in mammalian and avian species (Supplementary Fig. 6c). Therefore, we speculate that, like PDCoV, both HKU11 and HKU13 could use mammalian and avian APN as entry receptors. Unexpectedly, we have found that PDCoV shows a higher binding affinity to hAPN than to pAPN (Fig. 3e, Supplementary Table 2). Similar to our findings, a recent study reports that PDCoV exhibits stronger binding to chicken APN (cAPN) than to mouse APN (mAPN), feline APN (fAPN) or hAPN. Consistently, cAPN transfected cells support more efficient replication and production of PDCoV than cells in which mAPN, fAPN, or hAPN is overexpressed[33]. However, whether hAPN could support more efficient replication of PDCoV in human cells awaits further research. APN knockout pigs have been generated by different research groups[43,44]. Though porcine alveolar macrophages from APN knockout pigs show resistance to PDCoV infection, the gene-edited pigs are still susceptible to

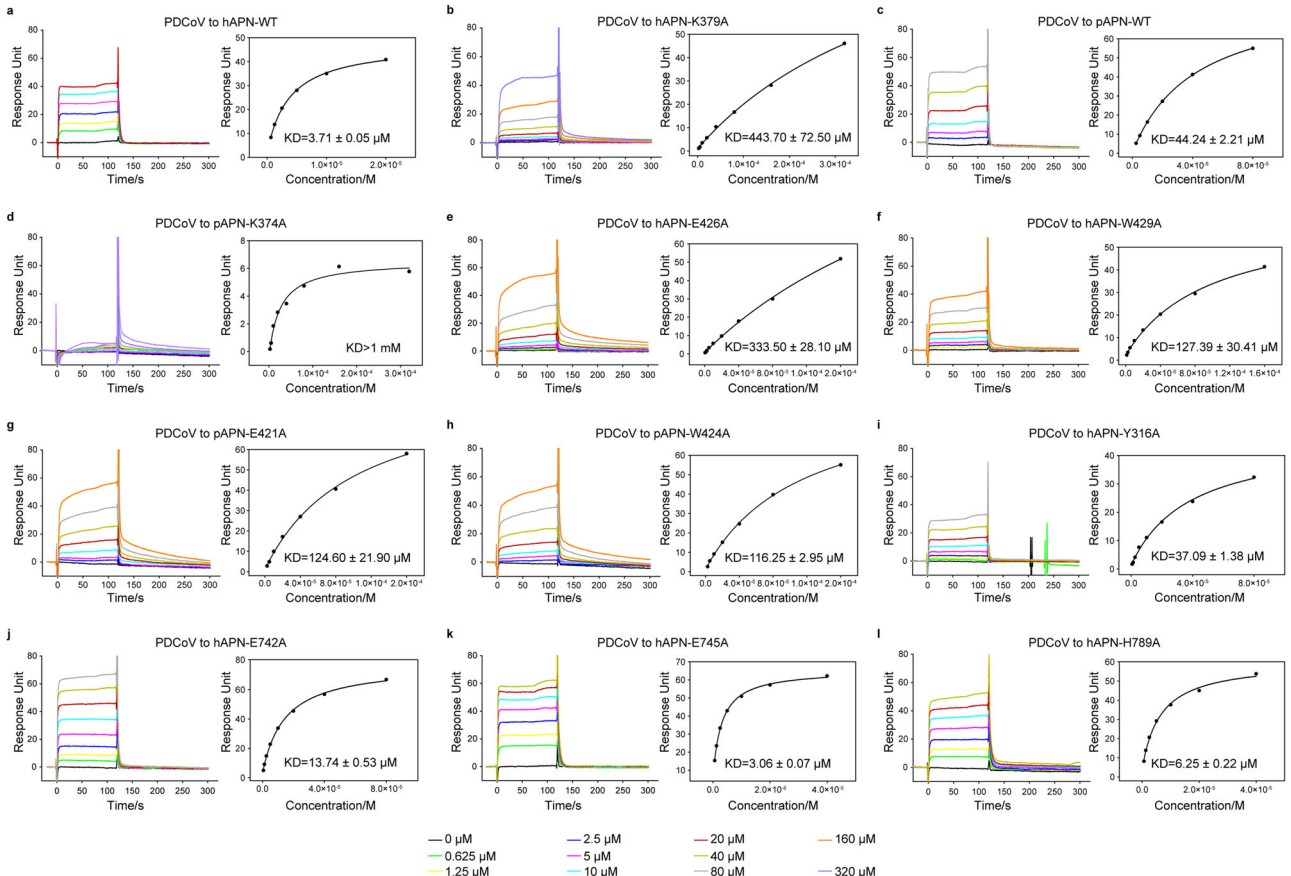

**Fig. 4 Binding of PDCoV RBD to wildtype (WT) or mutant hAPN/pAPN measured by SPR. a** Binding of PDCoV RBD to hAPN WT. **b**, **e**, **f**, **i–l** Binding of PDCoV RBD to different hAPN mutants. **c** Binding of PDCoV RBD to pAPN WT. **d**, **g**, **h** Binding of PDCoV RBD to different pAPN mutants. The equilibrium binding curve and derived dissociation constant is shown for each pair of interacting proteins. KD values are expressed as the mean ± SD, n = 2. SPR sensorgrams are provided as a Source Data file.

PDCoV[44]. Another group generated APN and CD163 double knockout (DKO) pigs and showed that DKO pigs were less susceptible to PDCoV infection, providing in vivo evidence of APN as one of the PDCoV receptors[45]. Meanwhile, as PDCoV is able to infect APN knockout pigs, there are probably other receptors that mediate PDCoV entry into cells. However, APN is important for PDCoV to establish infection. A recent study reveals that APN brings PDCoV to the endocytic pathway for later membrane fusion and genome release[46].

The mechanism of how alpha- and deltacoronaviruses bind to a common receptor reported here is different from that of HCoV-NL63 and SARS-CoV, which are alpha- and betacoronavirus, respectively[47,48]. HCoV-NL63 and SARS-CoV also share a common cellular receptor, ACE2[14,16]. Like PDCoV and HCoV-229E, the RBD of SARS-CoV and HCoV-NL63 also vary extensively, but they bind to a common region on ACE2[47,48]. Mutation of K353 and nearby residues on ACE2 decreases or abolishes the binding of both SARS-CoV and HCoV-NL63[49]. Instead, here we show that PDCoV, HCoV-229E, and PRCoV bind to different sites of the same receptor APN. Therefore, coronaviruses from different genera have evolved to target the same or distinct regions of a common receptor.

Previous studies indicate that free APN could adopt open, intermediate, and close conformations[50]. The difference among various conformations lies in the relative orientations of DII and DIV. In the open conformation, DII is away from DIV, which creates a channel to the active site of the enzyme, whereas in the close conformation, DII moves ~20 Å closer to DIV, leading to

limited accessibility of the active site[50]. It has been proposed that peptide substrates gain access to the active site of APN in the open conformation and undergo hydrolysis in the close conformation. After hydrolysis, the enzyme converts to open conformation again for substrate release[51]. PRCoV binds to APN in its open conformation whereas both HCoV-229E and PDCoV captured APN in the close conformation[21–23]. PRCoV, but not HCoV-229E binding site on APN differs when APN is in the open and close conformations[22]. PRCoV is a natural deletion variant of TGEV and the sequence identity between RBDs of these two viruses is 97%. Residues on PRCoV RBD that contact pAPN are 100% identical to their counterparts on TGEV RBD. This strongly suggests that TGEV and PRCoV bind to the same regions on pAPN[21]. Consistently, enzyme inhibitors trap APN in the close conformation inhibit TGEV, but not HCoV-229E infection[50]. In our study, PDCoV footprint engages both DII and DIV (Fig. 2a, b), which undergo large movement during the conversion of different APN conformations (Supplementary Fig. 2a–c). Based on our structural analysis, hydrogen bonding important for binding between PDCoV RBD and APN, including E320–W424 and F318–K379 will be disrupted when APN changes from close to open conformation (Supplementary Fig. 2d–f). As a result, enzyme inhibitors blocking the APN in the open formation would probably interfere with PDCoV infection. In a previous study[19], Li et al. constructed APN chimeric proteins via swapping DII (or DIV) of fAPN and hAPN. Based on the differential binding of PDCoV RBD to native and chimeric APN molecules expressed on HeLa cells, they concluded that APN DII

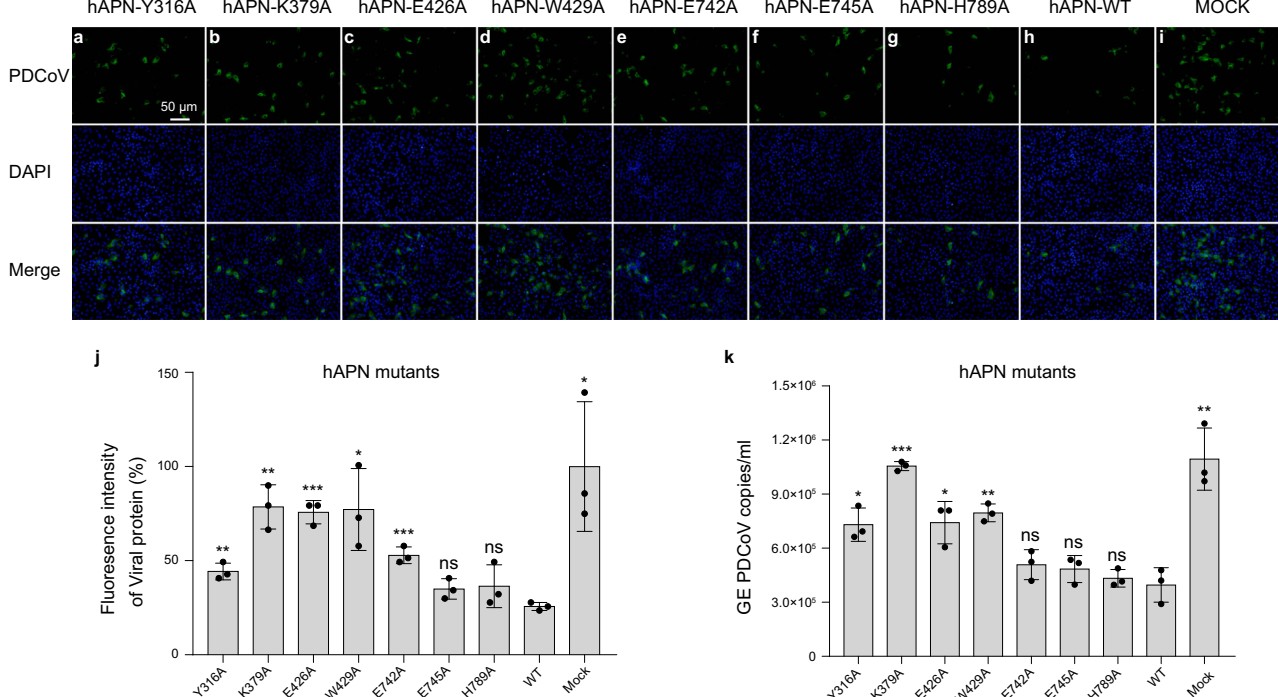

**Fig. 5 Y316, K379, E426, and W429 are the key residues on hAPN that inhibit PDCoV replication. a–i** PDCoV (MOI = 0.1) was preincubated with soluble wildtype or mutant hAPN proteins at a concentration of 80 μg/ml at 37 °C for 1 h, then the virus-APN protein mixtures were added onto a monolayer of LLC-PK1 cells in 24-well plates. After incubation at 4 °C for 1 h, the plates were washed twice with PBS and then cultured in DMEM containing 7.5 μg/ml trypsin at 37 °C with 5% CO$_2$. At 8 h post-infection (hpi), the samples were subject to immunofluorescence assay (IFA). Porcine polyclonal antibodies were used to detect PDCoV (green), and cell nuclei was stained with DAPI (blue). Scale bar, 50 μm. **j** The fluorescence intensity in **a–i** was determined by software Image J version 1.52i. Data are expressed as mean ± SD, n = 3. Error bars denote standard deviations for triplicate samples. An unpaired two-tailed $t$-test was used to determine the statistical significance. *$p < 0.05$; **$p < 0.01$; ***$p < 0.001$; ns no significance. **k** At 2 hpi, the samples from **a–i** were collected for RT-qPCR to determine the PDCoV genome equivalents (GE). Data are expressed as the mean ± SD, n = 3. Error bars denote standard deviations for triplicate samples. An unpaired two-tailed $t$-test was used to determine the statistical significance. *$p < 0.05$; **$p < 0.01$; ***$p < 0.001$; ns no significance. Values for fluorescence intensity and GE are provided as a Source Data file.

is a critical determinant for PDCoV. However, in addition to DII, we have visualized that a substantial portion of PDCoV RBD docks on DIV of APN in the structures. We postulate that the discrepancy may be caused by the different conformations adopted by chimeric and native APN proteins. In our structures, PDCoV RBD captured both hAPN and pAPN in close conformations. Swapping either of these two domains may therefore change the overall conformation of APN and thus affects its binding to PDCoV RBD.

The cryoEM structure of the PDCoV S protein ectodomain shows that in the prefusion stage, the RBD is in a "lying state", buried in the trimeric structure of S protein[52,53]. When we superimpose the PDCoV RBD–APN complex to the trimeric structure of PDCoV S protein ectodomain, severe clashes are observed between APN and RBD from neighboring S protein (Supplementary Fig. 7), suggesting that direct binding of APN to S protein in the "lying state" is highly impossible. Previous cryoEM studies captured coronavirus spike protein with RBD both in the "lying state" and "standing state"[54]. In the "standing state", the RBD is readily accessible for binding to receptors. Therefore, we propose that PDCoV RBD needs to transit from "lying state" to "standing state" to expose the surface for receptor binding.

As a potential zoonotic pathogen, PDCoV also poses risk to the human population[11]. The work reported here shows the detailed mechanism of how PDCoV binds to porcine and human APN receptors, indicating the risk of virus adaptation to the human population. This work also provides a valuable drug targets for controlling PDCoV infection.

## Methods

**Cell lines and virus**. Hi5 and sf9 insect cells were maintained in the SIM HF medium and SIM SF medium (Sino Biological Inc., Beijing, China) at 27 °C, respectively. LLC-PK1 cells and BHK-21 cells were cultured in Dulbecco's modified Eagle's medium (DMEM) supplemented with 5% and 10% fetal bovine serum (Tianhang Biotech, Hangzhou, China), respectively, at 37 °C with 5% CO$_2$.

PDCoV CZ2020 strain (GenBank accession No. OK546242) was isolated from a piglet suffering severe diarrhea and passaged in LLC-PK1 cells with DMEM supplemented with 7.5 μg/ml trypsin. The porcine anti-PDCoV hyperimmune serum was produced in the lab.

**Protein expression and purification**. The hAPN, pAPN, and PDCoV RBD proteins were expressed using the Bac-to-Bac baculovirus expression system (Invitrogen). The cDNA encoding residues 300–419 of the PDCoV S protein RBD (GenBank: AML40825.1) was codon-optimized for insect cells and synthesized by the Shanghai Generay Biotech. The coding sequences of hAPN ectodomain (GenBank: M22324.1, residues 62–963) and pAPN ectodomain (GenBank: MN514021.1, residues 58–961) were amplified by PCR from the Huh7 cells and pig liver tissue, respectively. A gp67 signal peptide sequence was added to the N-terminus of each construct to facilitate protein secretion. A 6×His tag was introduced at the C terminus of each construct to assist purification. Transfection and virus amplification was done in sf9 cells, and recombinant proteins were expressed in Hi5 cells. The target proteins, secreted in the Hi5 cell culture supernatants, were purified by Ni-NTA affinity chromatography (HisTrap™ FF), followed by purification with anion exchange chromatography (Resource™ Q column) and gel filtration (Superdex 200 Increase 10/300 GL column).

**Protein crystallization**. The RBD of the PDCoV S protein was mixed with the ectodomain of hAPN or pAPN at a molar ratio of 2:1 (RBD: hAPN or RBD: pAPN). The complex was concentrated to 6 mg/ml for crystallization trials. The crystals of the PDCoV RBD–hAPN complex were obtained via the hanging drop method by mixing 1 μl protein with 1 μl reservoir solution (25% w/v Polyethylene glycol 1500, 0.1 M BIS–TRIS propane pH 9.0, and 0.1 M Sodium chloride). For the

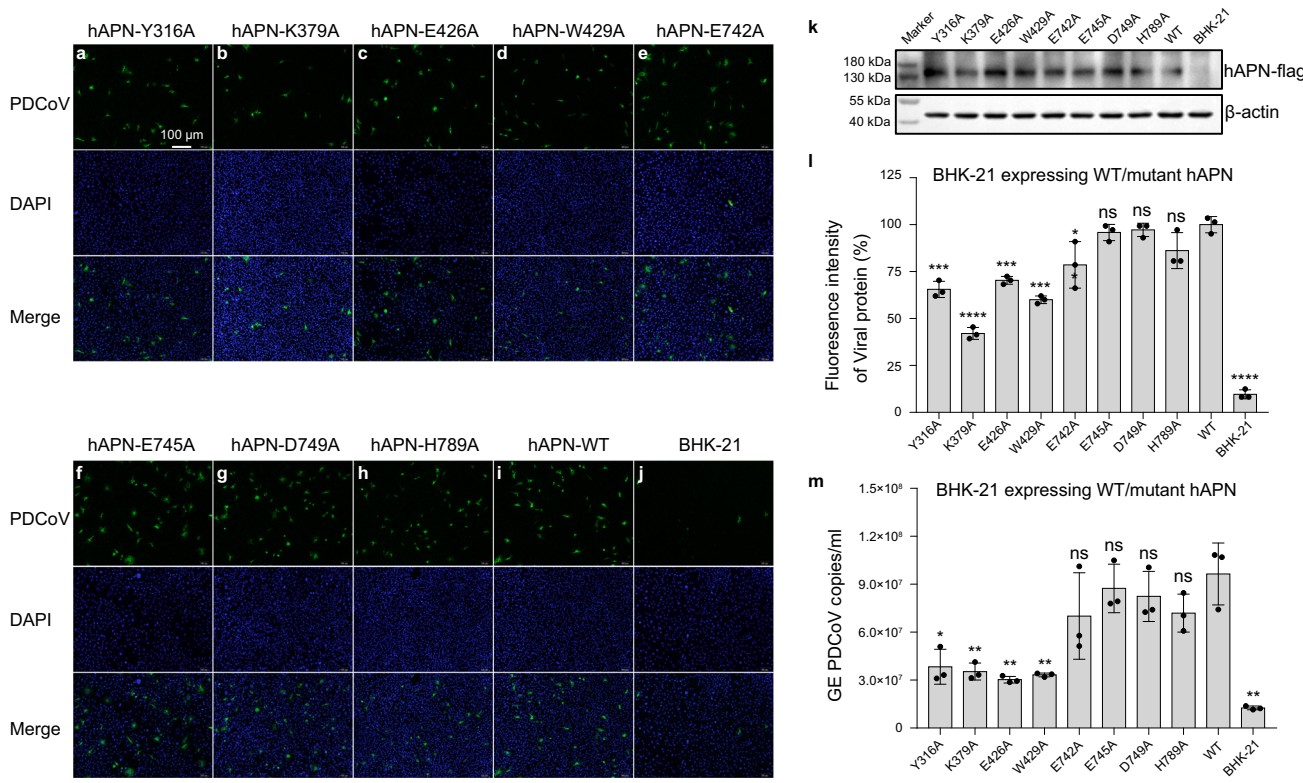

**Fig. 6 PDCoV infection was significantly reduced in BHK-21 cells overexpressing hAPN mutants (Y316A, K379A E426A, and W429A) compared to overexpression of wild-type hAPN. a–j** BHK-21 cell lines overexpressing wild-type or mutant hAPN proteins in 24-well plates were inoculated with 1 MOI of PDCoV CZ2020 strain. After incubation at 4 °C for 1 h, the plates were washed twice with PBS and then cultured in DMEM containing 0.5 μg/ml trypsin at 37 °C with 5% $CO_2$. At 8 hpi, the samples were subject to IFA. Porcine polyclonal antibodies were used to detect PDCoV (green), and cell nuclei was stained with DAPI (blue). Scale bar, 100 μm. **k** Cells from **a** to **j** were harvested with lysis buffer (50 mM Tris-HCl (pH 7.4), 150 mM NaCl, 1% NP-40, 0.1% SDS) supplemented with PMSF (Sigma-Aldrich). The cell lysates were boiled at 100 °C for 10 min then separated by SDS-PAGE. After electroblotting onto polyvinylidene difluoride membranes (Millipore), the membranes were blocked with 5% skimmed milk and then probed with indicated primary and secondary antibodies, and visualized using the chemiluminescent substrate. **l** The fluorescence intensity in **a–j** was determined by software Image J version 1.52i. Data are expressed as mean ± SD, $n = 3$. Error bars denote standard deviations for triplicate samples. An unpaired two-tailed $t$-test was used to determine the statistical significance. *$p < 0.05$; ***$p < 0.001$; ****$p < 0.0001$; ns no significance. **m** At 8 hpi, the samples from **a** to **j** were collected for RT-qPCR to determine the PDCoV genome equivalents (GE). Data are expressed as the mean ± SD, $n = 3$. Error bars denote standard deviations for triplicate samples. An unpaired two-tailed $t$-test was used to determine the statistical significance. *$p < 0.05$; **$p < 0.01$; ns no significance. Values for fluorescence intensity and GE are provided as a Source Data file.

PDCoV RBD–pAPN complex, crystals were grown in 8% w/v Polyethylene glycol 1000, 8% w/v polyethylene glycol 8000 and 20% w/v Glycerol.

**Data collection and structure determination.** The crystals were flash-frozen with liquid nitrogen in reservoir solutions supplemented with 20% glycerol. Diffraction data were collected at Shanghai Synchrotron Radiation Facility beamline BL18U1. All data were indexed, integrated, and scaled with XDS[55]. The structure was solved by molecular replacement in Phaser program[56], using hAPN (PDB code: 6atk) as the search model. The PDCoV RBD was traced unambiguously with Buccaneer[57] after density modification with Parrot[58]. Further rounds of iterative model building and refinement were performed using phenix.refine[59] and COOT[60], respectively. The stereochemical quality of the final model was assessed with the Molprobity[61]. Data processing and refinement statistics are summarized in Table 1. BSA of PDCoV RBD–hAPN and PDCoV RBD–pAPN complexes was calculated by the PISA program[62]. The structural figures were generated using PyMOL (http://www.pymol.org) or chimera[63].

**SPR assay.** SPR experiments were performed using the BIAcore X-100 system at room temperature. hAPN/pAPN WT or mutant proteins were immobilized on the CM5 sensor chips (GE Healthcare) via amine coupling. Serially diluted PDCoV RBD WT or mutant proteins in HEPES buffer (0.01 M HEPES, 0.15 M NaCl, 0.5% surfactant P20, 3 mM EDTA, pH 7.4) were followed over the chips. The kinetic data were analyzed with the Biacore X100 Evaluation software using the steady-state affinity model.

**Generation of stable BHK-21 cell lines expressing APN wildtype and mutant proteins.** APN wild-type or mutant genes was individually cloned into pCDNA 4.0

plasmid with a C-terminal Flag tag. The plasmids were used to transfect BHK-21 cells using Lipofectamine 2000 reagent (Invitrogen) according to the manufacturer's instructions. After incubation for 24 h, cells were treated with 300 μg/ml Zeocin™ (Invitrogen) for 1 week. After that, single colony cells were collected. Mock and APN transfected BHK-21 cell colonies were harvested for analysis of protein expression.

**Quantitative real-time PCR (RT-qPCR).** LLC-PK1 cells were seeded into 24-well plates and cultured at 37 °C with 5% $CO_2$. PDCoV (MOI = 0.1) was mixed with different concentrations of soluble wildtype or mutant pAPN/hAPN at 37 °C for 1 h, then added onto LLC-PK1 in 24-well plates. After incubation of the plates at 4 °C for 1 h, the plates were rinsed with PBS two times and then added with 500 μl of DMEM containing 7.5 μg/ml trypsin. Monolayers of BHK-21 cells stably express wild-type or mutant hAPN proteins in 24-well plates were inoculated with 1 MOI of PDCoV. After incubation at 4 °C for 1 h, the plates were washed twice with PBS and then cultured in DMEM supplemented with 0.5 μg/ml trypsin at 37 °C with 5% $CO_2$. PDCoV infected cells were subject to immunofluorescence assays (IFAs) at indicated time points.

Total RNA was extracted from cells infected with PDCoV at indicated time points using a HiPure RNA extraction kit (Magen Biotech, Shanghai, China). Then, the RNA was transcribed into cDNA using a reverse transcription kit (Vazyme Biotech, Nanjing, China). RT-qPCR experiments were performed at least in triplicate using SYBR green PCR master mix (Vazyme Biotech, Nanjing, China) in an ABI 7500 real-time PCR system (Applied Biosystems).

**Immunofluorescence and immunoblotting assays.** For IFA, cells infected with PDCoV in 24-well plates were washed with PBS 3 times, then fixed with 4%

paraformaldehyde for 15 min at room temperature. After washing three times with PBS, the cells were permeabilized with cold methanol for 10 min, followed by washing thrice with PBS and blocked with 5% skimmed milk for 1 h at 37 °C. Cells were rinsed with PBS three times, then incubated with pig anti-PDCoV primary antibody at 1:500 dilution for 1 h at 37 °C, followed by incubation with FITC-conjugated goat Anti-Pig IgG (Abcam, Cat: ab6911) at 1:1000 dilution for 1 h at 37 °C. Finally, cells were stained with 0.01% 4′,6-diamidino-2-phenylindole (DAPI), and washed three times. Fluorescent images were collected with a fluorescence microscope (Nikon).

For the immunoblotting assay, cells were harvested with lysis buffer (50 mM Tris-HCl (pH 7.4), 150 mM NaCl, 1% NP-40, 0.1% sodium dodecyl sulfate) supplemented with PMSF (Sigma-Aldrich). The cell lysates were boiled at 100 °C for 10 min and separated by sodium dodecyl sulfate-polyacrylamide gel electrophoresis. After electroblotting onto polyvinylidene difluoride membranes (Millipore), the membranes were blocked with 5% skimmed milk and then probed with indicated primary and secondary antibodies, and visualized using the chemiluminescent substrate. Primary antibodies are anti-Flag tag mouse mAb (Abmart, Cat: M20008S) and anti-β-actin mouse mAb (Proteintech, Cat: 66009-1-Ig) with a dilution of 1:5000. Secondary antibodies are HRP-conjugated goat anti-Mouse IgG(H + L) (Proteintech, Cat: SA00001-1) with a dilution of 1:2000.

**Adsorption assay**. LLC-PK1 cells cultured in 24-well plates were preincubated with pAPN (80 μg/ml) for 1 h at 37 °C and then prechilled at 4 °C for 1 h. The growth medium was then replaced with a mixture of pAPN (80 μg/ml) and PDCoV (MOI = 0.1). Cells were then incubated at 4 °C for 1 h in the presence of 7.5 μg/ml trypsin. After washing three times with prechilled PBS, PDCoV M protein gene levels were measured by RT-qPCR with primer set 5′-ATCGACCA-CATGGCTCCAA-3′ and 5′-CAGCTCTTGCCCATGTAGCTT-3′.

**Penetration assay**. LLC-PK1 cells cultured in 24-well plates were prechilled at 4 °C for 1 h and then incubated with PDCoV (MOI = 0.1) at 4 °C for 1 h. The virus-containing medium was replaced with a fresh medium containing pAPN (80 μg/ml) in the presence of 7.5 μg/ml trypsin. The temperature was raised to 37 °C for 3 h and then the cells were washed with PBS (pH 3). PDCoV M protein gene levels were assessed by RT-qPCR.

**Statistics and reproducibility**. Biochemical experiments were replicated at least two times. Virus infection assays were repeated three times. Confirmation of the wildtype and mutant APN protein expression in BHK-21 cells in Fig. 6k was replicated two times. Data are expressed as mean ± SD. Statistical analyses were performed by unpaired two-tailed $t$-test between groups using GraphPad Prism Software version 7.00. The $p$ value < 0.05 was considered statistically significant. The detailed statistical significance of differences for each experiment was provided in corresponding figure legends.

**Reporting summary**. Further information on research design is available in the Nature Research Reporting Summary linked to this article.

## Data availability

The data that support this study are available from the corresponding author upon reasonable request. Coordinates and structure factors of PDCoV RBD–hAPN complex and PDCoV RBD–pAPN complex are available in the Protein Data Bank under accession codes 7VPQ and 7VPP, respectively. Source data are provided with this paper.

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

## Acknowledgements

This work is supported by the National Key Research and Development Program of China (Grant no. 2021YFD1801104, B.L.), the Fundamental Research Funds for the Central Universities (Grant nos. KYRC2021007 and KJQN202150, S.Z., and KYQN2022013, Y.X.), National Natural Science Foundation of China (NSFC, Grant nos. 32170158 and 32000659, S.Z.; 32100128, Y.X.; 32130106, Z.W.), Natural Science Foundation of Jiangsu Province (Grant nos. BK20190003, B.L.; BK20200545SBK2020041541, S.Z.; BK20200553SBK2020041729, Y.X.), Jiangsu Agriculture Science and Technology Innovation Fund (Grant no. CX(20)3094, B.L.) and High-level personnel project of Jiangsu Province (Grant no. JSSCBS20210290, Y.X.). We thank staff of BL18U1 beamlines at Shanghai Synchrotron Radiation Facility for their technical assistance in crystal data collection.

## Author contributions

W.J., X.F., Z.L., and Y.L. expressed and purified the recombinant proteins. W.J. and C.X. carried out the SPR experiment. Q.P., S. Zhao, J.L. performed the PDCoV infection assay. Z.W. guided the generation of APN expressing stable cell lines. W.J. generated stable BHK-21 cell lines overexpressing APN wild-type and mutant proteins. Y.X., W.J., and Z.L. collected the X-ray diffraction data. S. Zhang solved the crystal structures. Q.P., B.L., and S. Zhang designed the experiments. Q.P., B.L., S. Zhang, J.W., Y.X., R.C., and G.M. analyzed the data and wrote the paper.

## Competing interests

The authors declare no competing interests.
