## [Peer Review File · Nature Communications]

Structures of a deltacoronavirus spike protein bound to porcine and human receptorsReviewers' Comments:

Reviewer #1:

Remarks to the Author:

Although individual structures of S protein of PDCoV (cryoEM) and human aminopeptidase N (hAPN) have been solved, the complex structure of S-APN has not been solved yet. In this manuscript, Ji et al solved the complex crystal structures of RBD of PDCoV with hAPN and porcine APN (pAPN), and found PDCoV RBD binds to a common region shared by hAPN and pAPN, which differs from the regions in respective APNs recognized by RBDs of 229E and PRCoV. They further identified several conserved residues in APN critical for RBD binding and virus infection. These findings are very important, given the discovery of recent zoonotic transmission of PDCoVs to children in Haiti. However, there are some issues needed to be addressed.

1. Overall buried surface area (BSA) in PDCoV RBD/hAPN is almost identical to that of PDCoV RBD/pAPN and almost twice of that of 229E RBD/hAPN, but the K_d for PDCoV RBD/hAPN (3.71 μM) is about 10-fold less than that of PDCoV/pAPN (44.24 μM) and about 9-fold higher than that 229E/hACE2 (0.43 μM , Wong et al Nat Comm 2017). Why there is such big discrepancy between BSA and K_d value?

2. Again, compared to K_d value for the most of known RBD-receptor interactions at nM range, PDCoV RBD/hAPN is at 3.7 μM and PDCoV RBD/pAPN is at 44.24 μM , which are considered as weak interactions. How could authors rule out the possibility of the structures resulting from crystal packing effect during crystallization process?

3. Why mutations of contacting residues in D4 regions have minimal effect on overall interaction although D4 region interactions account for more than 60% binding in BSA?

4. Again, according to Li et al (PNAS 2018), only D2 domain is critical for PDCoV binding, not D4, but in complex structure, D2 only account 38% BSA, while D4 account for over 60%. Why?

5. The inhibitory effect of soluble APN on PDCoV entry is very limited (Fig 5). Mutant APNs may be evaluated in BHK21 cells for their effect on PDCoV virus infection by overexpression in BHK21 cells, a strategy commonly used in PDCoV.

6. Some minor comments:

a) In M&M and text, amino acids 300-419 were used to make RBD, but in Fig 1, RBD was labeled at 302-422.

b) Line 54, "viral and cell membrane" should be "viral and cellular membrane"

Reviewer #2:

Remarks to the Author:

The manuscript titled "Structures of a deltacoronavirus spike protein bound to porcine and human receptors indicate the risk of virus adaptation to humans" describes the crystal structures of PDCoV RBD in complex with hAPN and pAPN respectively, to gain insights into the mechanism of PDCoV interspecies transmission. The authors first revealed that these two complex structures are highly similar and PDCoV RBD contacted common regions on hAPN and pAPN. By comparing the structures of PDCoV RBD-APN with previously solved structures of HCoV-229E RBD-APN and PRCoV RBD-APN, the authors clearly showed that PDCoV targeted very different regions on APN from α coronaviruses. The detailed biochemical and virus infection results further convincingly validated that PDCoV bound to conserved residues from homologous APN receptors to infect cells.

This is the first structural study of a deltacoronavirus RBD bound to cellular receptors. The study is novel and timely, as amid the COVID-19 pandemic, cross species transmission of coronavirus is of great interest to the world. Specially related to this study, Lednicky et al recently reported infection of PDCoV among Haitian children (Lednicky et al., Nature, 2021). The structures and other data reported in this study explicitly demonstrate that PDCoV targets conserved sites on hAPN and pAPN, thus providing the likely mechanism for PDCoV zoonotic transmission. The results are convincing and the

manuscript is well-written with in-depth discussion.

There are some minor issues that should be addressed before publication.

1. The authors should include homology analyses of PDCoV strain used in this study to the strains that have been reported to infect human (Lednicky et al., Nature, 2021).
2. Related to 1, the authors should compare the RBD regions from typical PDCoV strains isolated from different countries (Lines 43-44), especially the human-infecting ones reported recently. This may provide key information on PDCoV zoonotic transmission, as has been shown for SARS-CoV.
3. Lednicky et al (Nature, 2021) identified different mutations in S1 subunit of Hu-PDCoV (PDCoV isolated from human) strains. Are these mutations related to this study? The authors should clarify this.
4. Cite the recent study on human infection of PDCoV (Lednicky et al., Nature, 2021).
5. In Figure 2, the authors compared the sequences of hAPN and pAPN, but did not include the overall sequence identity of these two proteins. They should add this information in the manuscript.
6. The authors should indicate the program to calculate buried surface area on PDCoV RBD or APN in the method section and cite the corresponding reference.
7. The shape complementarity value between the interacting interfaces of APN and RBD should be included. This parameter could be calculated by sc program in CCP4.

Reviewer #3:

Remarks to the Author:

This paper describes the x-ray crystal structures of the porcine deltacoronavirus (PDCoV) receptor binding domain (RBD) in complex with porcine APN (pAPN) and human APN (hAPN).

The structures show that the two complexes are essentially identical and that the RBD binds at site that is well conserved between pAPN and hAPN. This observation provides a structural rationalization for the ability of this virus to infect multiple hosts including pigs and humans. As such, the work is of considerable significance given the health concerns associated with the cross-species transmission of new coronaviruses to humans.

Interestingly, the site on APN bound by PDCoV differs from the sites at which the alphacoronaviruses, human HCoV-229E and porcine transmissible gastroenteritis virus, bind to APN (ie. 3 sites on APN now identified). This observation provides further evidence of how coronaviruses have on several occasions evolved to exploit this receptor. Again, a concern regarding the potential for new coronavirus threats capable of human-to-human transmission.

As part of their characterization, the authors perform mutagenesis and binding studies to probe the significance of selected residues on both the RBD and APN. The BIAcore binding curves look reasonable but the theoretical fit to the raw data needs to be shown in all cases. The equilibrium binding curves used to derive the K_d values must also be shown and the residuals to the fit reported along with the K_d values. These curves could be put into the supplementary data section.

The authors also show that both soluble hAPN and pAPN inhibit infection by PDCoV in a cell-based assay, an indication that APN is the major if not the only receptor/factor required for entry. Repeating these experiments with mutant forms of APN (the same ones tested in the binding studies) does not provide additional information on what residues are involved in "entry" but rather just reflects the

weakened affinity for the RBD. That said, they provide an additional measure of the role played by selected amino acids in the RBD-APN interaction.

Based on the data collection and refinement statistics, and the validation reports, the x-ray structures appear to be of reasonable quality given their resolution. At more than once place in the text, it must be indicated that the RMSD values reported are in angstroms (eg. line 121: 0.63 Å not 0.63).

Reviewer #1:

Although individual structures of S protein of PDCoV (cryoEM) and human aminopeptidase N (hAPN) have been solved, the complex structure of S-APN has not been solved yet. In this manuscript, Ji et al solved the complex crystal structures of RBD of PDCoV with hAPN and porcine APN (pAPN), and found PDCoV RBD binds to a common region shared by hAPN and pAPN, which differs from the regions in respective APNs recognized by RBDs of 229E and PRCoV. They further identified several conserved residues in APN critical for RBD binding and virus infection. These findings are very important, given the discovery of recent zoonotic transmission of PDCoVs to children in Haiti. However, there are some issues needed to be addressed.

We thank reviewer #1 for the positive comments.

1. Overall buried surface area (BSA) in PDCoV RBD/hAPN is almost identical to that of PDCoV RBD/pAPN and almost twice of that of 229E RBD/hAPN, but the K_d for PDCoV RBD/hAPN (3.71 μM) is about 10-fold less than that of PDCoV/pAPN (44.24 μM) and about 9-fold higher than that 229E/hACE2 (0.43 μM , Wong et al Nat Comm 2017). Why there is such big discrepancy between BSA and K_d value?

As the reviewer has pointed out, the buried surface area (BSA) is related to binding affinity (measured as K_d)¹. However, as far as we know, there is no strict correlation between BSA and K_d ². In addition to BSA, other factors, including electrostatics, non-interacting interfaces, conformational changes, etc., also contribute substantially to binding affinity². Electrostatic interactions were found experimentally and computationally to be important for protein binding³. Consistently, polar and charged residues on the protein-protein interfaces are “hot spots” that affect protein binding affinity³. Therefore, we compared the electrostatic potential, which reflected the distribution of polar and charged residues, of PDCoV RBD-hAPN, PDCoV RBD-pAPN and HCoV-229E RBD-hAPN. As shown in the figure below (Supplementary Fig. 1d, e in the revised manuscript), there is only partial charge complementarity between PDCoV RBD and hAPN/pAPN. However, the RBMs on HCoV-229E RBD is fully positively charged and docks onto the totally negatively charged counterpart on hAPN. The stronger binding between hAPN and HCoV-229E RBD may thus result from complete charge complementarity between them.

Since the electric potential distribution in pAPN and hAPN is highly similar, this unlikely accounts for the 10-fold difference in affinity of PDCoV to hAPN and pAPN. However, PDCoV RBD shows better shape complementarity for hAPN than pAPN (Sc value 0.66 vs 0.56), which may explain the stronger interaction between PDCoV RBD and hAPN. In addition, the sequence identity between pAPN and hAPN is about 79% and therefore other residues at the non-interacting interface may also lead to the difference in affinity.

Now we have included discussion about this in the revised manuscript (Lines 239-254):

“Although the BSA at the interface of PDCoV RBD-hAPN or PDCoV RBD-pAPN is larger than that of HCoV-229E RBD-hAPN, the binding affinity of PDCoV RBD to hAPN or pAPN is lower compared to that of HCoV-229E RBD to hAPN. Electrostatic interactions were found experimentally and computationally to be important for protein binding³⁹. Therefore, we compared the electrostatic potential, calculated using APBS⁴⁰ and PDB2PQR⁴¹ packages, of PDCoV RBD-hAPN, PDCoV RBD-pAPN and HCoV-229E RBD-hAPN (Supplementary Fig. 1d, e). There is only partial charge complementarity between PDCoV RBD and hAPN/pAPN. However, the RBMs on HCoV-229E RBD is fully positively charged and docks onto the negatively charged counterpart on hAPN. The stronger binding between hAPN and HCoV-229E RBD may thus result from complete charge complementarity between them. Since the electric potential distribution in pAPN and hAPN is highly similar, this unlikely accounts for the 10-fold difference in their affinity to PDCoV RBD. However, PDCoV RBD shows better shape complementarity for hAPN than pAPN (Sc value 0.66 vs 0.56), which may explain the stronger interaction between PDCoV RBD and hAPN. In addition, the sequence identity

between pAPN and hAPN is 79% and therefore other residues at the non-interacting interface may also lead to the difference in affinity.”

2. Again, compared to K_d value for the most of known RBD-receptor interactions at nM range, PDCoV RBD/hAPN is at 3.7 uM and PDCoV RBD/pAPN is at 44.24 uM, which are considered as weak interactions. How could authors rule out the possibility of the structures resulting from crystal packing effect during crystallization process?

We thank the reviewer for raising this concern. As the reviewer has indicated, the interaction of PDCoV RBD to APN is not strong. However, the heterocomplexes reported in our study are very unlikely to be the result of crystal packing for the reasons mentioned below.

(1) The interfaces between PDCoV RBD and hAPN/pAPN are present in the asymmetric units of the structures and are not generated via crystallographic symmetry operations (translation and rotation). The figure below shows the packing of molecules in a unit cell (with RBD colored in blue, and hAPN/pAPN colored in cyan, purple, orange and green), which reveals that the crystal packing interfaces are very different from the RBD-APN interacting interfaces in the asymmetric units (circled with red dotted line).

(2) We have observed very similar interacting interfaces on PDCoV RBD-hAPN and PDCoV RBD-pAPN complexes, even though they crystallized in two space groups (C222 and P41212) that display very different patterns of crystal packing.

(3) Based on structure guided mutagenesis, we have mapped residues on PDCoV RBD-hAPN or PDCoV RBD-pAPN interface that affect binding of respective proteins via SPR assay (Fig. 4 and Supplementary Fig. 3 and 4). Our virus infection experiment has further revealed that mutation of residues on the interface significantly affect PDCoV infection of host cells as well (Fig. 6 a-j). The proteins of these assays are not in crystalline form, thus ruling out the possibility of crystal packing.

3. Why mutations of contacting residues in D4 regions have minimal effect on overall interaction although D4 region interactions account for more than 60% binding in BSA?

There are in total 15 residues on hAPN DIV that contact PDCoV RBD (Supplementary Table 1). 6 out of these 15 residues are involved in hydrogen bonding or salt bridge interactions, while most of the contacts are mediated by either hydrophobic or Van der Waals (vdW) interactions. Although the sum of many hydrophobic or vdW interactions contribute significantly to protein-protein binding, each individual interaction is quite weak. Therefore, in our study, we tried to disrupt hydrogen bonding and salt bridges formed between APN and PDCoV RBD. 3 of those 6 residues on APN DIV are hydrogen bonded with PDCoV RBD via their backbone but not side chain atoms. Mutation of these residues is unlikely to change the backbone conformation and therefore would not be expected to disrupt interaction of respective proteins.

Specifically, R741, E742, I743, E745, D749 and H789, on hAPN DIV interact with PDCoV RBD via hydrogen bonds or salt bridges. Since only the backbone atoms of R741 and I743 are involved in hydrogen bonding, we choose not to mutate these two residues. Different from R741 and I743, E745 forms hydrogen bonds with PDCoV RBD via both its backbone and side chain atoms (Fig. 3c). Accordingly, E745A mutation would only break hydrogen bonding that involves its side chain atoms. We think this may explain why E745A mutation has little effects on its binding to PDCoV RBD. While H789A mutation on APN DIV only slightly decreases binding, mutation of Y398 on PDCoV RBD, which is hydrogen bonded with H789, dramatically reduces binding, indicating that H789 on DIV contributes to the interaction of respective proteins. Mutation of E742 on hAPN DIV did significantly decrease PDCoV RBD binding to APN.

4. Again, according to Li et al (PNAS 2018), only D2 domain is critical for PDCoV binding, not D4, but in complex structure, D2 only account 38% BSA, while D4 account for over 60%. Why?

We thank the reviewer for pointing out the discrepancy between our work and a previous study done by Li et al⁴. In Li et al's study, they constructed APN chimeric proteins via swapping DII (or DIV) of feline APN (fAPN) and hAPN. Based on the differential binding of PDCoV RBD to native and chimeric APN molecules expressed on HeLa cells, they concluded that APN DII was a critical determinant for PDCoV. However, in addition to APN DII, we have visualized that a substantial portion of PDCoV RBD docks onto APN DIV in our structures. We postulate that the discrepancy may be caused by the different conformations adopted by chimeric and native APN proteins. In our structures, PDCoV RBD captured both hAPN and pAPN in close conformations. The interaction of DII and DIV has been shown to decrease dramatically when APN changes from close to open conformation⁵. Swapping either of these two domains may therefore change the overall conformation of APN and thus affects its binding to PDCoV RBD. This has been shown for TGEV, which binds to APN in open but not close conformation⁵.

Additionally, PDCoV could infect HeLa cells overexpressing fAPN, which was shown in figure 6 of the 2018 PNAS paper. Therefore, fAPN itself is not a totally "clean" negative control for domain swapping. Li et al's work thus could not rule out DIV is involved in binding to PDCoV RBD, as they put in the paper "However, it should be noted that, because fAPN is bound to low affinity by PDCoV S1 and can be used as an entry receptor by the virus (*vide infra*), no results can be drawn from any negative results obtained in this assay".

Now we have updated discussion section of the manuscript to address this discrepancy (Lines 321-330):

"In a previous study¹⁹, Li et al. constructed APN chimeric proteins via swapping DII (or DIV) of fAPN and hAPN. Based on the differential binding of PDCoV RBD to native and chimeric APN molecules expressed on HeLa cells, they concluded that APN DII is a critical determinant for PDCoV. However, in addition to DII, we have visualized that a substantial portion of PDCoV RBD docks on DIV of APN in the structures. We postulate that the discrepancy may be caused by the different conformations adopted by chimeric and native APN proteins. In our structures, PDCoV RBD captured both hAPN and pAPN in close

conformations. Swapping either of these two domains may therefore change the overall conformation of APN and thus affects its binding to PDCoV RBD.”

5. The inhibitory effect of soluble APN on PDCoV entry is very limited (Fig 5). Mutant APNs may be evaluated in BHK21 cells for their effect on PDCoV virus infection by overexpression in BHK21 cells, a strategy commonly used in PDCoV.

As the reviewer has suggested, we have generated BHK-21 cell lines stably express wildtype and mutant APN proteins. The ability of PDCoV to infect these cell lines are evaluated via IFA and RT-qPCR. The figure below (Fig. 6 in the revised manuscript) shows that mutation of Y316, K379, E426 and W429 on hAPN significantly decrease PDCoV infection, which further confirms that these residues are important for virus binding and infection.

Now we have revised the result section of the manuscript to include the data (Lines 216-220).

“Overexpression of hAPN mutants in BHK-21 cells reduced PDCoV infection in varying degrees compared to overexpression of hAPN wildtype protein (Fig. 6a-k). Consistent with the result from protein blocking assay, Y316A, K379A, E426A or W429A led to ~70% reduction of infection (Fig. 6l, m), indicating these residues play important roles in virus receptor binding and infection”

6. Some minor comments:

a) In M&M and text, amino acids 300-419 were used to make RBD, but in Fig 1, RBD was labeled at 302-422.

We have changed the label on revised Fig. 1 as shown below.

b) Line 54, “viral and cell membrane” should be “viral and cellular membrane”

The typo has been corrected (Line 57 in the revised manuscript) and thank the reviewer for pointing this out.

“Binding of S1 subunit to viral receptors leads to large conformational change in S2 subunit that drives fusion of viral and cellular membranes¹³”

Reviewer #2:

The manuscript titled "Structures of a deltacoronavirus spike protein bound to porcine and human receptors indicate the risk of virus adaptation to humans" describes the crystal structures of PDCoV RBD in complex with hAPN and pAPN respectively, to gain insights into the mechanism of PDCoV interspecies transmission. The authors first revealed that these two complex structures are highly similar and PDCoV RBD contacted common regions on hAPN and pAPN. By comparing the structures of PDCoV RBD-APN with previously solved structures of HCoV-229E RBD-APN and PRCoV RBD-APN, the authors clearly showed that PDCoV targeted very different regions on APN from α coronaviruses. The detailed biochemical and virus infection results further convincingly validated that PDCoV bound to conserved residues from homologous APN receptors to infect cells.

This is the first structural study of a deltacoronavirus RBD bound to cellular receptors. The study is novel and timely, as amid the COVID-19 pandemic, cross species transmission of coronavirus is of great interest to the world. Specially related to this study, Lednicky et al recently reported infection of PDCoV among Hatian children (Lednicky et al., Nature, 2021). The structures and other data reported in this study explicitly demonstrate that PDCoV targets conserved sites on hAPN and pAPN, thus providing the likely mechanism for PDCoV zoonotic transmission. The results are convincing and the manuscript is well-written with in-depth discussion.

We thank reviewer #2 for the positive comments.

1. The authors should include homology analyses of PDCoV strain used in this study to the strains that have been reported to infect human (Lednicky et al., Nature, 2021).

The sequence identity between CZ2020 (from this study) and strains infecting human ranges from 98.3 to 98.6%. As the reviewer suggested, now we have included a supplementary figure (Supplementary Fig. 6a) in the revised manuscript to compare the RBDs of S protein from different PDCoV strains.

	310	320	330	340	350	360	370	380	390	400	410	419
Haiti/Human/0081-4/2014	PKLPELEVQLN	SAHMDPGEARLDSVT	INGNTSYCVTK	YFRLETFNFMCTGCTMNLRTDTC	SPDL	SAVNNGMSFSQFCLSTESGACEMKII	IVTVVWNYLLRQRLYVTV	AV	EGQHTGTTS			
Haiti/Human/0256-12015	PKLPELEVQLN	SAHMDPGEARLDSVT	INGNTSYCVTK	YFRLETFNFMCTGCTMNLRTDTC	SPDL	SAVNNGMSFSQFCLSTESGACEMKII	IVTVVWNYLLRQRLYVTV	AV	EGQHTGTTS			
CZ2020	PKLPELEVQLN	SAHMDPGEARLDSVT	INGNTSYCVTK	YFRLETFNFMCTGCTMNLRTDTC	SPDL	SAVNNGMSFSQFCLSTESGACEMKII	IVTVVWNYLLRQRLYVTV	AV	EGQHTGTTS			
USA/Ohio445/2014	PKLPELEVQLN	SAHMDPGEARLDSVT	INGNTSYCVTK	YFRLETFNFMCTGCTMNLRTDTC	SPDL	SAVNNGMSFSQFCLSTESGACEMKII	IVTVVWNYLLRQRLYVTV	AV	EGQHTGTTS			
YMG/JPN/2014	PKLPELEVQLN	SAHMDPGEARLDSVT	INGNTSYCVTK	YFRLETFNFMCTGCTMNLRTDTC	SPDL	SAVNNGMSFSQFCLSTESGACEMKII	IVTVVWNYLLRQRLYVTV	AV	EGQHTGTTS			
KNU16-11	PKLPELEVQLN	SAHMDPGEARLDSVT	INGNTSYCVTK	YFRLETFNFMCTGCTMNLRTDTC	SPDL	SAVNNGMSFSQFCLSTESGACEMKII	IVTVVWNYLLRQRLYVTV	AV	EGQHTGTTS			
PDCoV/CHGD/2016	PKLPELEVQLN	SAHMDPGEARLDSVT	INGNTSYCVTK	YFRLETFNFMCTGCTMNLRTDTC	SPDL	SAVNNGMSFSQFCLSTESGACEMKII	IVTVVWNYLLRQRLYVTV	AV	EGQHTGTTS			
PDCoV/CHJXN/2015	PKLPELEVQLN	SAHMDPGEARLDSVT	INGNTSYCVTK	YFRLETFNFMCTGCTMNLRTDTC	SPDL	SAVNNGMSFSQFCLSTESGACEMKII	IVTVVWNYLLRQRLYVTV	AV	EGQHTGTTS			
NH	PKLPELEVQLN	SAHMDPGEARLDSVT	INGNTSYCVTK	YFRLETFNFMCTGCTMNLRTDTC	SPDL	SAVNNGMSFSQFCLSTESGACEMKII	IVTVVWNYLLRQRLYVTV	AV	EGQHTGTTS			
CHN-JS-2017	PKLPELEVQLN	SAHMDPGEARLDSVT	INGNTSYCVTK	YFRLETFNFMCTGCTMNLRTDTC	SPDL	SAVNNGMSFSQFCLSTESGACEMKII	IVTVVWNYLLRQRLYVTV	AV	EGQHTGTTS			
CHN-AH-2004	PKLPELEVQLN	SAHMDPGEARLDSVT	INGNTSYCVTK	YFRLETFNFMCTGCTMNLRTDTC	SPDL	SAVNNGMSFSQFCLSTESGACEMKII	IVTVVWNYLLRQRLYVTV	AV	EGQHTGTTS			
Vietnam/Binh21/2015	PKLPELEVQLN	SAHMDPGEARLDSVT	INGNTSYCVTK	YFRLETFNFMCTGCTMNLRTDTC	SPDL	SAVNNGMSFSQFCLSTESGACEMKII	IVTVVWNYLLRQRLYVTV	AV	EGQHTGTTS			
PDCoV/2016/Lao	PKLPELEVQLN	SAHMDPGEARLDSVT	INGNTSYCVTK	YFRLETFNFMCTGCTMNLRTDTC	SPDL	SAVNNGMSFSQFCLSTESGACEMKII	IVTVVWNYLLRQRLYVTV	AV	EGQHTGTTS			
Thailand/IS015L/2015	PKLPELEVQLN	SAHMDPGEARLDSVT	INGNTSYCVTK	YFRLETFNFMCTGCTMNLRTDTC	SPDL	SAVNNGMSFSQFCLSTESGACEMKII	IVTVVWNYLLRQRLYVTV	AV	EGQHTGTTS			

2. Related to 1, the authors should compare the RBD regions from typical PDCoV strains isolated from different countries (Lines 43-44), especially the human-infecting ones reported recently. This may provide key information on PDCoV zoonotic transmission, as has been shown for SARS-CoV.

We thank the reviewer for the above suggestion. As shown in the figure included in response to question 1 (Supplementary Fig. 6a in the revised manuscript), the sequence identity of RBDs of different PDCoV strains, including the human infecting ones, ranges from 96.1 to 100%. The RBM residues are strictly conserved, suggesting these strains may all share the capacity to infect different hosts.

We have we have updated the discussion section of the manuscript to address question 1 and 2 (Lines 256-261).

“PDCoV strains isolated from children in Haiti were highly similar to the pig strains detected in China and America. Specifically, the sequence identity of RBDs of different PDCoV strains, including the human infecting ones, ranges from 96.1 to 100%. The RBM residues of these PDCoV strains are strictly conserved (Supplementary Fig. 6a). This indicates the risk of cross-species transmission of PDCoV between pig and human population.”

3. Lednicky et al (Nature, 2021) identified different mutations in S1 subunit of Hu-PDCoV (PDCoV isolated from human) strains. Are these mutations related to this study? The authors should clarify this.

The S1 mutations identified in the above paper (Lednicky et al Nature, 2021) include residue 38 and 550, none of which is on the RBD. Therefore, in addition to RBD, other residues in S protein could also affect the host range of PDCoV.

We have updated the discussion section to include the above information (Lines 261-263).

“Lednicky et al. found two mutations outside RBD of S1 subunit of the human infecting PDCoV strain¹¹, suggesting other regions on S protein may also affect host range of PDCoV as well.”

4. Cite the recent study on human infection of PDCoV (Lednicky et al., Nature, 2021).

As the reviewer suggested, we have updated the references to include this paper (Reference 11).

“Lednicky, J. A. *et al.* Independent infections of porcine deltacoronavirus among Haitian children. *Nature* **600**, 133-137 (2021).”

5. In Figure 2, the authors compared the sequences of hAPN and pAPN, but did not include the overall sequence identity of these two proteins. They should add this information in the manuscript.

The overall sequence identity between hAPN and pAPN is 79%, which now we have included in the result section of the revised manuscript (Line 197).

“hAPN and pAPN share 79% amino acid sequence identity. Consistent with the similar PDCoV binding modes of hAPN and pAPN (Fig. 2a, b), most of the residues (19/25) contacting PDCoV RBD are strictly conserved between hAPN and pAPN (Fig. 2c)”

6. The authors should indicate the program to calculate buried surface area on PDCoV RBD or APN in the method section and cite the corresponding reference.

We are sorry for missing this information. We used PISA program to calculate the buried surface area at the interface between different protein complexes, including PDCoV RBD-pAPN, PDCoV RBD-hAPN and HCoV-229E RBD-hAPN. Now we have update the method section (Lines 385-386) and reference list.

“BSA of PDCoV RBD-hAPN and PDCoV RBD-pAPN complexes was calculated by the PISA program⁶²,”

“Krissinel, E. & Henrick, K. Inference of Macromolecular Assemblies from Crystalline State. *Journal of Molecular Biology* **372**, 774-797 (2007)”

7. The shape complementarity value between the interacting interfaces of APN and RBD should be included. This parameter could be calculated by sc program in CCP4.

We have calculated the shape complementarity values between PDCoV RBD/hAPN and PDCoV RBD/pAPN, which are 0.66 and 0.56 respectively. The shape complementarity value now is included in the result section of the revised manuscript (Lines 116-118).

“The shape complementarity values, calculated with Sc³⁷, are 0.66 and 0.56 for PDCoV RBD-hAPN and PDCoV RBD-pAPN, suggesting PDCoV shows better complementarity for hAPN in terms of shape.”

Reviewer #3:

This paper describes the x-ray crystal structures of the porcine deltacoronavirus (PDCoV) receptor binding domain (RBD) in complex with porcine APN (pAPN) and human APN (hAPN).

The structures show that the two complexes are essentially identical and that the RBD binds at site that is well conserved between pAPN and hAPN. This observation provides a structural

rationalization for the ability of this virus to infect multiple hosts including pigs and humans. As such, the work is of considerable significance given the health concerns associated with the cross-species transmission of new coronaviruses to humans.

Interestingly, the site on APN bound by PDCoV differs from the sites at which the alphacoronviruses, human HCoV-229E and porcine transmissible gastroenteritis virus, bind to APN (ie. 3 sites on APN now identified). This observation provides further evidence of how coronaviruses have on several occasions evolved to exploit this receptor. Again, a concern regarding the potential for new coronavirus threats capable of human-to-human transmission.

We thank reviewer #3 for the positive comments.

(1) As part of their characterization, the authors perform mutagenesis and binding studies to probe the significance of selected residues on both the RBD and APN. The BIAcore binding curves look reasonable but the theoretical fit to the raw data needs to be shown in all cases. The equilibrium binding curves used to derive the Kd values must also be shown and the residuals to the fit reported along with the Kd values. These curves could be put into the supplementary data section.

As the reviewer suggested, now we have added equilibrium binding curves in revised Fig.4 and also in Supplementary Fig. 3 and 4. Since we have used the steady state instead of the kinetical model to derive the binding affinity, the theoretic fits to the raw data are not available. The residual standard deviation now is included in Supplementary table 2 and 3.

Fig. 4 Binding of PDCoV RBD to wildtype (WT) or mutant hAPN / pAPN measured by SPR

Supplementary Fig. 3 Binding of WT or mutant PDCoV RBD to hAPN measured by SPR

Supplementary Fig. 4 Binding of WT or mutant PDCoV RBD to pAPN measured by SPR

Supplementary Table 2 The affinities and residual standard deviation (SE) of PDCoV (WT) binding to pAPN/hAPN (mutant)

	PDCoV-WT	
	KD(M)	SE
hAPN-Y316A	3.571E-05	4.3E-06
	3.846E-05	5.2E-06
hAPN-E426A	3.054E-04	4.6E-05
	3.616E-04	4.9E-05
hAPN-W429A	1.578E-04	2.4E-05
	9.698E-05	1.1E-05
hAPN-K379A	3.712E-04	2.5E-05
	5.162E-04	6.9E-05
hAPN-E742A	1.427E-05	1.1E-06
	1.321E-05	1.1E-06
pAPN-E421A	1.027E-04	1.2E-05
	1.465E-04	1.4E-05
pAPN-W424A	1.133E-04	8.7E-06
	1.192E-04	8.3E-06
pAPN-K374A	2.516E-05	7.3E-06
	1.820E-07	1.5E-06

Supplementary Table 3 The affinities and residual standard deviation SE of pAPN/hAPN (WT) binding to PDCoV RBD (WT/mutant)

	hAPN		pAPN	
	KD(M)	SE	KD (M)	SE
PDCoV-WT	3.763E-06	2.5E-07	4.203E-05	9.9E-07
	3.662E-06	3.0E-07	4.645E-05	7.2E-07
PDCoV-F318A	1.119E-04	3.8E-05	8.171E-04	7.3E-04
	1.539E-04	9.9E-05	5.113E-04	4.4E-04
PDCoV-E320A	1.347E-04	2.6E-05	2.790E-04	3.0E-05
	1.881E-04	3.9E-05	2.195E-04	2.2E-05
PDCoV-R322A	2.557E-04	2.6E-05	4.544E-04	6.4E-05
	3.350E-04	6.8E-05	5.530E-04	9.1E-05
PDCoV-R357A	1.993E-04	5.7E-05	2.765E-05	2.7E-06
	2.319E-04	5.3E-05	6.052E-05	9.8E-06
PDCoV-W396A	2.339	26	45.85	3.7E+03
	0.088	0.61	44.27	3.2E+03
PDCoV-N397A	3.347E-06	4.6E-07	3.402E-05	1.3E-06
	3.625E-06	6.7E-07	5.378E-05	1.4E-05
PDCoV-Y398A	3.091E-04	7.8E-05	2.574	10
	7.304E-04	2.6E-04	1.112	31

(2) The authors also show that both soluble hAPN and pAPN inhibit infection by PDCoV in a cell-based assay, an indication that APN is the major if not the only receptor/factor required for entry. Repeating these experiments with mutant forms of APN (the same ones tested in the binding studies) does not provide additional information on what residues are involved in "entry" but rather just reflects the weakened affinity for the RBD. That said, they provide an additional measure of the role played by selected amino acids in the RBD-APN interaction.

We thank the reviewer for raising this concern. Now we have included data of PDCoV infecting BHK-21 cells overexpress wildtype and mutant APN proteins, which prove that the key residues on PDCoV RBD directly affect virus binding and infection (Fig. 6a-k).

We have included this result in the revised manuscript (Lines 216-220):

“Overexpression of hAPN mutants in BHK-21 cells reduced PDCoV infection in varying degrees compared to overexpression of hAPN wildtype protein (Fig. 6a-k). Consistent with the result from protein blocking assay, Y316A, K379A, E426A or W429A led to ~70% reduction of infection (Fig. 6l, m), indicating these residues play important roles in virus receptor binding and infection”

Based on the data collection and refinement statistics, and the validation reports, the x-ray structures appear to be of reasonable quality given their resolution.

(3) At more than once place in the text, it must be indicated that the RMSD values reported are in angstroms (eg. line 121: 0.63 Å not 0.63).

We are sorry for missing the unit information of the RMSD, which is now included in the revised manuscript (Lines 114-116, 610-612).

“The structures of the two complexes exhibit high similarity, with a root mean square deviation (RMSD) of 0.63 Å over 976 equivalent C α atoms”

“The two structures exhibit high similarity and the root mean square deviation (RMSD) between them is 0.63 Å”

References:

- 1 Horton, N. & Lewis, M. Calculation of the free energy of association for protein complexes. *Protein Sci.* **1**, 169-181 (1992).
- 2 Vangone, A. & Bonvin, A. M. J. J. Contacts-based prediction of binding affinity in protein–protein complexes. *eLife* **4**, e07454 (2015).
- 3 Kundrotas, P. J. & Alexov, E. Electrostatic Properties of Protein-Protein Complexes. *Biophys. J.* **91**, 1724-1736 (2006).
- 4 Li, W. *et al.* Broad receptor engagement of an emerging global coronavirus may potentiate its diverse cross-species transmissibility. *Proc. Natl. Acad. Sci. U. S. A.* **115**, E5135 (2018).
- 5 Santiago, C. *et al.* Allosteric inhibition of aminopeptidase N functions related to tumor growth and virus infection. *Sci. Rep.* **7**, 46045 (2017).

Reviewers' Comments:

Reviewer #1:

Remarks to the Author:

This reviewer is satisfied with the responses and changes.

Reviewer #2:

Remarks to the Author:

All look good at me!

Reviewer #3:

Remarks to the Author:

I have read the rebuttal and revised manuscript and find that my questions/concerns have been addressed.

In response to reviewer 1's concern that the complexes observed might be crystal packing artefacts, the authors suggest that the complexes are correct (one piece of evidence) since the sites of interaction are contained within the asymmetric unit. This is not correct - crystal packing artefacts are not restricted to symmetry related contacts. However, the observation that the complexes are found in two different space groups is good evidence that they are not artefacts. Moreover, the mutagenesis work (RBD and APN) also strongly supports the validity of the complexes as the authors argue.

Minor points:

- 1) line 102 "The PDCoV RBD was located" should be "The PDCoV RBD is located"
- 2) line 288 "These researches also indicate" needs to be re-written

Reviewer #1:

Although individual structures of S protein of PDCoV (cryoEM) and human aminopeptidase N (hAPN) have been solved, the complex structure of S-APN has not been solved yet. In this manuscript, Ji et al solved the complex crystal structures of RBD of PDCoV with hAPN and porcine APN (pAPN), and found PDCoV RBD binds to a common region shared by hAPN and pAPN, which differs from the regions in respective APNs recognized by RBDs of 229E and PRCoV. They further identified several conserved residues in APN critical for RBD binding and virus infection. These findings are very important, given the discovery of recent zoonotic transmission of PDCoVs to children in Haiti. However, there are some issues needed to be addressed.

We thank reviewer #1 for the positive comments.

1. Overall buried surface area (BSA) in PDCoV RBD/hAPN is almost identical to that of PDCoV RBD/pAPN and almost twice of that of 229E RBD/hAPN, but the K_d for PDCoV RBD/hAPN (3.71 μM) is about 10-fold less than that of PDCoV/pAPN (44.24 μM) and about 9-fold higher than that 229E/hACE2 (0.43 μM , Wong et al Nat Comm 2017). Why there is such big discrepancy between BSA and K_d value?

As the reviewer has pointed out, the buried surface area (BSA) is related to binding affinity (measured as K_d)¹. However, as far as we know, there is no strict correlation between BSA and K_d ². In addition to BSA, other factors, including electrostatics, non-interacting interfaces, conformational changes, etc., also contribute substantially to binding affinity². Electrostatic interactions were found experimentally and computationally to be important for protein binding³. Consistently, polar and charged residues on the protein-protein interfaces are “hot spots” that affect protein binding affinity³. Therefore, we compared the electrostatic potential, which reflected the distribution of polar and charged residues, of PDCoV RBD-hAPN, PDCoV RBD-pAPN and HCoV-229E RBD-hAPN. As shown in the figure below (Supplementary Fig. 1d, e in the revised manuscript), there is only partial charge complementarity between PDCoV RBD and hAPN/pAPN. However, the RBMs on HCoV-229E RBD is fully positively charged and docks onto the totally negatively charged counterpart on hAPN. The stronger binding between hAPN and HCoV-229E RBD may thus result from complete charge complementarity between them.

Since the electric potential distribution in pAPN and hAPN is highly similar, this unlikely accounts for the 10-fold difference in affinity of PDCoV to hAPN and pAPN. However, PDCoV RBD shows better shape complementarity for hAPN than pAPN (Sc value 0.66 vs 0.56), which may explain the stronger interaction between PDCoV RBD and hAPN. In addition, the sequence identity between pAPN and hAPN is about 79% and therefore other residues at the non-interacting interface may also lead to the difference in affinity.

Now we have included discussion about this in the revised manuscript (Lines 239-254):

“Although the BSA at the interface of PDCoV RBD-hAPN or PDCoV RBD-pAPN is larger than that of HCoV-229E RBD-hAPN, the binding affinity of PDCoV RBD to hAPN or pAPN is lower compared to that of HCoV-229E RBD to hAPN. Electrostatic interactions were found experimentally and computationally to be important for protein binding³⁹. Therefore, we compared the electrostatic potential, calculated using APBS⁴⁰ and PDB2PQR⁴¹ packages, of PDCoV RBD-hAPN, PDCoV RBD-pAPN and HCoV-229E RBD-hAPN (Supplementary Fig. 1d, e). There is only partial charge complementarity between PDCoV RBD and hAPN/pAPN. However, the RBMs on HCoV-229E RBD is fully positively charged and docks onto the negatively charged counterpart on hAPN. The stronger binding between hAPN and HCoV-229E RBD may thus result from complete charge complementarity between them. Since the electric potential distribution in pAPN and hAPN is highly similar, this unlikely accounts for the 10-fold difference in their affinity to PDCoV RBD. However, PDCoV RBD shows better shape complementarity for hAPN than pAPN (Sc value 0.66 vs 0.56), which may explain the stronger interaction between PDCoV RBD and hAPN. In addition, the sequence identity

between pAPN and hAPN is 79% and therefore other residues at the non-interacting interface may also lead to the difference in affinity.”

2. Again, compared to K_d value for the most of known RBD-receptor interactions at nM range, PDCoV RBD/hAPN is at 3.7 μ M and PDCoV RBD/pAPN is at 44.24 μ M, which are considered as weak interactions. How could authors rule out the possibility of the structures resulting from crystal packing effect during crystallization process?

We thank the reviewer for raising this concern. As the reviewer has indicated, the interaction of PDCoV RBD to APN is not strong. However, the heterocomplexes reported in our study are very unlikely to be the result of crystal packing for the reasons mentioned below.

(1) The interfaces between PDCoV RBD and hAPN/pAPN are present in the asymmetric units of the structures and are not generated via crystallographic symmetry operations (translation and rotation). The figure below shows the packing of molecules in a unit cell (with RBD colored in blue, and hAPN/pAPN colored in cyan, purple, orange and green), which reveals that the crystal packing interfaces are very different from the RBD-APN interacting interfaces in the asymmetric units (circled with red dotted line).

(2) We have observed very similar interacting interfaces on PDCoV RBD-hAPN and PDCoV RBD-pAPN complexes, even though they crystallized in two space groups (C222 and P41212) that display very different patterns of crystal packing.

(3) Based on structure guided mutagenesis, we have mapped residues on PDCoV RBD-hAPN or PDCoV RBD-pAPN interface that affect binding of respective proteins via SPR assay (Fig. 4 and Supplementary Fig. 3 and 4). Our virus infection experiment has further revealed that mutation of residues on the interface significantly affect PDCoV infection of host cells as well (Fig. 6 a-j). The proteins of these assays are not in crystalline form, thus ruling out the possibility of crystal packing.

3. Why mutations of contacting residues in D4 regions have minimal effect on overall interaction although D4 region interactions account for more than 60% binding in BSA?

There are in total 15 residues on hAPN DIV that contact PDCoV RBD (Supplementary Table 1). 6 out of these 15 residues are involved in hydrogen bonding or salt bridge interactions, while most of the contacts are mediated by either hydrophobic or Van der Waals (vdW) interactions. Although the sum of many hydrophobic or vdW interactions contribute significantly to protein-protein binding, each individual interaction is quite weak. Therefore, in our study, we tried to disrupt hydrogen bonding and salt bridges formed between APN and PDCoV RBD. 3 of those 6 residues on APN DIV are hydrogen bonded with PDCoV RBD via their backbone but not side chain atoms. Mutation of these residues is unlikely to change the backbone conformation and therefore would not be expected to disrupt interaction of respective proteins.

Specifically, R741, E742, I743, E745, D749 and H789, on hAPN DIV interact with PDCoV RBD via hydrogen bonds or salt bridges. Since only the backbone atoms of R741 and I743 are involved in hydrogen bonding, we choose not to mutate these two residues. Different from R741 and I743, E745 forms hydrogen bonds with PDCoV RBD via both its backbone and side chain atoms (Fig. 3c). Accordingly, E745A mutation would only break hydrogen bonding that involves its side chain atoms. We think this may explain why E745A mutation has little effects on its binding to PDCoV RBD. While H789A mutation on APN DIV only slightly decreases binding, mutation of Y398 on PDCoV RBD, which is hydrogen bonded with H789, dramatically reduces binding, indicating that H789 on DIV contributes to the interaction of respective proteins. Mutation of E742 on hAPN DIV did significantly decrease PDCoV RBD binding to APN.

4. Again, according to Li et al (PNAS 2018), only D2 domain is critical for PDCoV binding, not D4, but in complex structure, D2 only account 38% BSA, while D4 account for over 60%. Why?

We thank the reviewer for pointing out the discrepancy between our work and a previous study done by Li et al⁴. In Li et al's study, they constructed APN chimeric proteins via swapping DII (or DIV) of feline APN (fAPN) and hAPN. Based on the differential binding of PDCoV RBD to native and chimeric APN molecules expressed on HeLa cells, they concluded that APN DII was a critical determinant for PDCoV. However, in addition to APN DII, we have visualized that a substantial portion of PDCoV RBD docks onto APN DIV in our structures. We postulate that the discrepancy may be caused by the different conformations adopted by chimeric and native APN proteins. In our structures, PDCoV RBD captured both hAPN and pAPN in close conformations. The interaction of DII and DIV has been shown to decrease dramatically when APN changes from close to open conformation⁵. Swapping either of these two domains may therefore change the overall conformation of APN and thus affects its binding to PDCoV RBD. This has been shown for TGEV, which binds to APN in open but not close conformation⁵.

Additionally, PDCoV could infect HeLa cells overexpressing fAPN, which was shown in figure 6 of the 2018 PNAS paper. Therefore, fAPN itself is not a totally "clean" negative control for domain swapping. Li et al's work thus could not rule out DIV is involved in binding to PDCoV RBD, as they put in the paper "However, it should be noted that, because fAPN is bound to low affinity by PDCoV S1 and can be used as an entry receptor by the virus (*vide infra*), no results can be drawn from any negative results obtained in this assay".

Now we have updated discussion section of the manuscript to address this discrepancy (Lines 321-330):

"In a previous study¹⁹, Li et al. constructed APN chimeric proteins via swapping DII (or DIV) of fAPN and hAPN. Based on the differential binding of PDCoV RBD to native and chimeric APN molecules expressed on HeLa cells, they concluded that APN DII is a critical determinant for PDCoV. However, in addition to DII, we have visualized that a substantial portion of PDCoV RBD docks on DIV of APN in the structures. We postulate that the discrepancy may be caused by the different conformations adopted by chimeric and native APN proteins. In our structures, PDCoV RBD captured both hAPN and pAPN in close

conformations. Swapping either of these two domains may therefore change the overall conformation of APN and thus affects its binding to PDCoV RBD.”

5. The inhibitory effect of soluble APN on PDCoV entry is very limited (Fig 5). Mutant APNs may be evaluated in BHK21 cells for their effect on PDCoV virus infection by overexpression in BHK21 cells, a strategy commonly used in PDCoV.

As the reviewer has suggested, we have generated BHK-21 cell lines stably express wildtype and mutant APN proteins. The ability of PDCoV to infect these cell lines are evaluated via IFA and RT-qPCR. The figure below (Fig. 6 in the revised manuscript) shows that mutation of Y316, K379, E426 and W429 on hAPN significantly decrease PDCoV infection, which further confirms that these residues are important for virus binding and infection.

Now we have revised the result section of the manuscript to include the data (Lines 216-220).

“Overexpression of hAPN mutants in BHK-21 cells reduced PDCoV infection in varying degrees compared to overexpression of hAPN wildtype protein (Fig. 6a-k). Consistent with the result from protein blocking assay, Y316A, K379A, E426A or W429A led to ~70% reduction of infection (Fig. 6l, m), indicating these residues play important roles in virus receptor binding and infection”

6. Some minor comments:

a) In M&M and text, amino acids 300-419 were used to make RBD, but in Fig 1, RBD was labeled at 302-422.

We have changed the label on revised Fig. 1 as shown below.

b) Line 54, “viral and cell membrane” should be “viral and cellular membrane”

The typo has been corrected (Line 57 in the revised manuscript) and thank the reviewer for pointing this out.

“Binding of S1 subunit to viral receptors leads to large conformational change in S2 subunit that drives fusion of viral and cellular membranes¹³”

Reviewer #2:

The manuscript titled "Structures of a deltacoronavirus spike protein bound to porcine and human receptors indicate the risk of virus adaptation to humans" describes the crystal structures of PDCoV RBD in complex with hAPN and pAPN respectively, to gain insights into the mechanism of PDCoV interspecies transmission. The authors first revealed that these two complex structures are highly similar and PDCoV RBD contacted common regions on hAPN and pAPN. By comparing the structures of PDCoV RBD-APN with previously solved structures of HCoV-229E RBD-APN and PRCoV RBD-APN, the authors clearly showed that PDCoV targeted very different regions on APN from α coronaviruses. The detailed biochemical and virus infection results further convincingly validated that PDCoV bound to conserved residues from homologous APN receptors to infect cells.

This is the first structural study of a deltacoronavirus RBD bound to cellular receptors. The study is novel and timely, as amid the COVID-19 pandemic, cross species transmission of coronavirus is of great interest to the world. Specially related to this study, Lednicky et al recently reported infection of PDCoV among Hatian children (Lednicky et al., Nature, 2021). The structures and other data reported in this study explicitly demonstrate that PDCoV targets conserved sites on hAPN and pAPN, thus providing the likely mechanism for PDCoV zoonotic transmission. The results are convincing and the manuscript is well-written with in-depth discussion.

We thank reviewer #2 for the positive comments.

1. The authors should include homology analyses of PDCoV strain used in this study to the strains that have been reported to infect human (Lednicky et al., Nature, 2021).

The sequence identity between CZ2020 (from this study) and strains infecting human ranges from 98.3 to 98.6%. As the reviewer suggested, now we have included a supplementary figure (Supplementary Fig. 6a) in the revised manuscript to compare the RBDs of S protein from different PDCoV strains.

	310	320	330	340	350	360	370	380	390	400	410	419
Haiti/Human/0081-4/2014	PKLPELEVQLN	SAHMDPGEARLDSVT	INGNTSYCVTK	YFRLETFNFMCTGCTMNLRTDTC	SPDL	SAVNNGMSFSQFCLSTESGACEMKII	IVTVVWNYLLRQRL	YVTV	AV	EGQ	TH	TGTT
Haiti/Human/0256-12015	PKLPELEVQLN	SAHMDPGEARLDSVT	INGNTSYCVTK	YFRLETFNFMCTGCTMNLRTDTC	SPDL	SAVNNGMSFSQFCLSTESGACEMKII	IVTVVWNYLLRQRL	YVTV	AV	EGQ	TH	TGTT
CZ2020	PKLPELEVQLN	SAHMDPGEARLDSVT	INGNTSYCVTK	YFRLETFNFMCTGCTMNLRTDTC	SPDL	SAVNNGMSFSQFCLSTESGACEMKII	IVTVVWNYLLRQRL	YVTV	AV	EGQ	TH	TGTT
USA/Ohio445/2014	PKLPELEVQLN	SAHMDPGEARLDSVT	INGNTSYCVTK	YFRLETFNFMCTGCTMNLRTDTC	SPDL	SAVNNGMSFSQFCLSTESGACEMKII	IVTVVWNYLLRQRL	YVTV	AV	EGQ	TH	TGTT
YMG/JPN2014	PKLPELEVQLN	SAHMDPGEARLDSVT	INGNTSYCVTK	YFRLETFNFMCTGCTMNLRTDTC	SPDL	SAVNNGMSFSQFCLSTESGACEMKII	IVTVVWNYLLRQRL	YVTV	AV	EGQ	TH	TGTT
KNU16-11	PKLPELEVQLN	SAHMDPGEARLDSVT	INGNTSYCVTK	YFRLETFNFMCTGCTMNLRTDTC	SPDL	SAVNNGMSFSQFCLSTESGACEMKII	IVTVVWNYLLRQRL	YVTV	AV	EGQ	TH	TGTT
PDCoV/CHGD/2016	PKLPELEVQLN	SAHMDPGEARLDSVT	INGNTSYCVTK	YFRLETFNFMCTGCTMNLRTDTC	SPDL	SAVNNGMSFSQFCLSTESGACEMKII	IVTVVWNYLLRQRL	YVTV	AV	EGQ	TH	TGTT
PDCoV/CHJXNI2/2015	PKLPELEVQLN	SAHMDPGEARLDSVT	INGNTSYCVTK	YFRLETFNFMCTGCTMNLRTDTC	SPDL	SAVNNGMSFSQFCLSTESGACEMKII	IVTVVWNYLLRQRL	YVTV	AV	EGQ	TH	TGTT
NH	PKLPELEVQLN	SAHMDPGEARLDSVT	INGNTSYCVTK	YFRLETFNFMCTGCTMNLRTDTC	SPDL	SAVNNGMSFSQFCLSTESGACEMKII	IVTVVWNYLLRQRL	YVTV	AV	EGQ	TH	TGTT
CHN-JS-2017	PKLPELEVQLN	SAHMDPGEARLDSVT	INGNTSYCVTK	YFRLETFNFMCTGCTMNLRTDTC	SPDL	SAVNNGMSFSQFCLSTESGACEMKII	IVTVVWNYLLRQRL	YVTV	AV	EGQ	TH	TGTT
CHN-AH-2004	PKLPELEVQLN	SAHMDPGEARLDSVT	INGNTSYCVTK	YFRLETFNFMCTGCTMNLRTDTC	SPDL	SAVNNGMSFSQFCLSTESGACEMKII	IVTVVWNYLLRQRL	YVTV	AV	EGQ	TH	TGTT
Vietnam/Binh21/2015	PKLPELEVQLN	SAHMDPGEARLDSVT	INGNTSYCVTK	YFRLETFNFMCTGCTMNLRTDTC	SPDL	SAVNNGMSFSQFCLSTESGACEMKII	IVTVVWNYLLRQRL	YVTV	AV	EGQ	TH	TGTT
PDCoV/2016/Lao	PKLPELEVQLN	SAHMDPGEARLDSVT	INGNTSYCVTK	YFRLETFNFMCTGCTMNLRTDTC	SPDL	SAVNNGMSFSQFCLSTESGACEMKII	IVTVVWNYLLRQRL	YVTV	AV	EGQ	TH	TGTT
Thailand/IS015L/2015	PKLPELEVQLN	SAHMDPGEARLDSVT	INGNTSYCVTK	YFRLETFNFMCTGCTMNLRTDTC	SPDL	SAVNNGMSFSQFCLSTESGACEMKII	IVTVVWNYLLRQRL	YVTV	AV	EGQ	TH	TGTT

2. Related to 1, the authors should compare the RBD regions from typical PDCoV strains isolated from different countries (Lines 43-44), especially the human-infecting ones reported recently. This may provide key information on PDCoV zoonotic transmission, as has been shown for SARS-CoV.

We thank the reviewer for the above suggestion. As shown in the figure included in response to question 1 (Supplementary Fig. 6a in the revised manuscript), the sequence identity of RBDs of different PDCoV strains, including the human infecting ones, ranges from 96.1 to 100%. The RBM residues are strictly conserved, suggesting these strains may all share the capacity to infect different hosts.

We have we have updated the discussion section of the manuscript to address question 1 and 2 (Lines 256-261).

“PDCoV strains isolated from children in Haiti were highly similar to the pig strains detected in China and America. Specifically, the sequence identity of RBDs of different PDCoV strains, including the human infecting ones, ranges from 96.1 to 100%. The RBM residues of these PDCoV strains are strictly conserved (Supplementary Fig. 6a). This indicates the risk of cross-species transmission of PDCoV between pig and human population.”

3. Lednicky et al (Nature, 2021) identified different mutations in S1 subunit of Hu-PDCoV (PDCoV isolated from human) strains. Are these mutations related to this study? The authors should clarify this.

The S1 mutations identified in the above paper (Lednicky et al Nature, 2021) include residue 38 and 550, none of which is on the RBD. Therefore, in addition to RBD, other residues in S protein could also affect the host range of PDCoV.

We have updated the discussion section to include the above information (Lines 261-263).

“Lednicky et al. found two mutations outside RBD of S1 subunit of the human infecting PDCoV strain¹¹, suggesting other regions on S protein may also affect host range of PDCoV as well.”

4. Cite the recent study on human infection of PDCoV (Lednicky et al., Nature, 2021).

As the reviewer suggested, we have updated the references to include this paper (Reference 11).

“Lednicky, J. A. *et al.* Independent infections of porcine deltacoronavirus among Haitian children. *Nature* **600**, 133-137 (2021).”

5. In Figure 2, the authors compared the sequences of hAPN and pAPN, but did not include the overall sequence identity of these two proteins. They should add this information in the manuscript.

The overall sequence identity between hAPN and pAPN is 79%, which now we have included in the result section of the revised manuscript (Line 197).

“hAPN and pAPN share 79% amino acid sequence identity. Consistent with the similar PDCoV binding modes of hAPN and pAPN (Fig. 2a, b), most of the residues (19/25) contacting PDCoV RBD are strictly conserved between hAPN and pAPN (Fig. 2c)”

6. The authors should indicate the program to calculate buried surface area on PDCoV RBD or APN in the method section and cite the corresponding reference.

We are sorry for missing this information. We used PISA program to calculate the buried surface area at the interface between different protein complexes, including PDCoV RBD-pAPN, PDCoV RBD-hAPN and HCoV-229E RBD-hAPN. Now we have update the method section (Lines 385-386) and reference list.

“BSA of PDCoV RBD-hAPN and PDCoV RBD-pAPN complexes was calculated by the PISA program⁶²,”

“Krissinel, E. & Henrick, K. Inference of Macromolecular Assemblies from Crystalline State. *Journal of Molecular Biology* 372, 774-797 (2007)”

7. The shape complementarity value between the interacting interfaces of APN and RBD should be included. This parameter could be calculated by sc program in CCP4.

We have calculated the shape complementarity values between PDCoV RBD/hAPN and PDCoV RBD/pAPN, which are 0.66 and 0.56 respectively. The shape complementarity value now is included in the result section of the revised manuscript (Lines 116-118).

“The shape complementarity values, calculated with Sc³⁷, are 0.66 and 0.56 for PDCoV RBD-hAPN and PDCoV RBD-pAPN, suggesting PDCoV shows better complementarity for hAPN in terms of shape.”

Reviewer #3:

This paper describes the x-ray crystal structures of the porcine deltacoronavirus (PDCoV) receptor binding domain (RBD) in complex with porcine APN (pAPN) and human APN (hAPN).

The structures show that the two complexes are essentially identical and that the RBD binds at site that is well conserved between pAPN and hAPN. This observation provides a structural

rationalization for the ability of this virus to infect multiple hosts including pigs and humans. As such, the work is of considerable significance given the health concerns associated with the cross-species transmission of new coronaviruses to humans.

Interestingly, the site on APN bound by PDCoV differs from the sites at which the alphacoronviruses, human HCoV-229E and porcine transmissible gastroenteritis virus, bind to APN (ie. 3 sites on APN now identified). This observation provides further evidence of how coronaviruses have on several occasions evolved to exploit this receptor. Again, a concern regarding the potential for new coronavirus threats capable of human-to-human transmission.

We thank reviewer #3 for the positive comments.

(1) As part of their characterization, the authors perform mutagenesis and binding studies to probe the significance of selected residues on both the RBD and APN. The BIAcore binding curves look reasonable but the theoretical fit to the raw data needs to be shown in all cases. The equilibrium binding curves used to derive the Kd values must also be shown and the residuals to the fit reported along with the Kd values. These curves could be put into the supplementary data section.

As the reviewer suggested, now we have added equilibrium binding curves in revised Fig.4 and also in Supplementary Fig. 3 and 4. Since we have used the steady state instead of the kinetical model to derive the binding affinity, the theoretic fits to the raw data are not available. The residual standard deviation now is included in Supplementary table 2 and 3.

Fig. 4 Binding of PDCoV RBD to wildtype (WT) or mutant hAPN / pAPN measured by SPR

Supplementary Fig. 3 Binding of WT or mutant PDCoV RBD to hAPN measured by SPR

Supplementary Fig. 4 Binding of WT or mutant PDCoV RBD to pAPN measured by SPR

Supplementary Table 2 The affinities and residual standard deviation (SE) of PDCoV (WT) binding to pAPN/hAPN (mutant)

	PDCoV-WT	
	KD(M)	SE
hAPN-Y316A	3.571E-05	4.3E-06
	3.846E-05	5.2E-06
hAPN-E426A	3.054E-04	4.6E-05
	3.616E-04	4.9E-05
hAPN-W429A	1.578E-04	2.4E-05
	9.698E-05	1.1E-05
hAPN-K379A	3.712E-04	2.5E-05
	5.162E-04	6.9E-05
hAPN-E742A	1.427E-05	1.1E-06
	1.321E-05	1.1E-06
pAPN-E421A	1.027E-04	1.2E-05
	1.465E-04	1.4E-05
pAPN-W424A	1.133E-04	8.7E-06
	1.192E-04	8.3E-06
pAPN-K374A	2.516E-05	7.3E-06
	1.820E-07	1.5E-06

Supplementary Table 3 The affinities and residual standard deviation SE of pAPN/hAPN (WT) binding to PDCoV RBD (WT/mutant)

	hAPN		pAPN	
	KD(M)	SE	KD (M)	SE
PDCoV-WT	3.763E-06	2.5E-07	4.203E-05	9.9E-07
	3.662E-06	3.0E-07	4.645E-05	7.2E-07
PDCoV-F318A	1.119E-04	3.8E-05	8.171E-04	7.3E-04
	1.539E-04	9.9E-05	5.113E-04	4.4E-04
PDCoV-E320A	1.347E-04	2.6E-05	2.790E-04	3.0E-05
	1.881E-04	3.9E-05	2.195E-04	2.2E-05
PDCoV-R322A	2.557E-04	2.6E-05	4.544E-04	6.4E-05
	3.350E-04	6.8E-05	5.530E-04	9.1E-05
PDCoV-R357A	1.993E-04	5.7E-05	2.765E-05	2.7E-06
	2.319E-04	5.3E-05	6.052E-05	9.8E-06
PDCoV-W396A	2.339	26	45.85	3.7E+03
	0.088	0.61	44.27	3.2E+03
PDCoV-N397A	3.347E-06	4.6E-07	3.402E-05	1.3E-06
	3.625E-06	6.7E-07	5.378E-05	1.4E-05
PDCoV-Y398A	3.091E-04	7.8E-05	2.574	10
	7.304E-04	2.6E-04	1.112	31

(2) The authors also show that both soluble hAPN and pAPN inhibit infection by PDCoV in a cell-based assay, an indication that APN is the major if not the only receptor/factor required for entry. Repeating these experiments with mutant forms of APN (the same ones tested in the binding studies) does not provide additional information on what residues are involved in "entry" but rather just reflects the weakened affinity for the RBD. That said, they provide an additional measure of the role played by selected amino acids in the RBD-APN interaction.

We thank the reviewer for raising this concern. Now we have included data of PDCoV infecting BHK-21 cells overexpress wildtype and mutant APN proteins, which prove that the key residues on PDCoV RBD directly affect virus binding and infection (Fig. 6a-k).

We have included this result in the revised manuscript (Lines 216-220):

“Overexpression of hAPN mutants in BHK-21 cells reduced PDCoV infection in varying degrees compared to overexpression of hAPN wildtype protein (Fig. 6a-k). Consistent with the result from protein blocking assay, Y316A, K379A, E426A or W429A led to ~70% reduction of infection (Fig. 6l, m), indicating these residues play important roles in virus receptor binding and infection”

Based on the data collection and refinement statistics, and the validation reports, the x-ray structures appear to be of reasonable quality given their resolution.

(3) At more than once place in the text, it must be indicated that the RMSD values reported are in angstroms (eg. line 121: 0.63 Å not 0.63).

We are sorry for missing the unit information of the RMSD, which is now included in the revised manuscript (Lines 114-116, 610-612).

“The structures of the two complexes exhibit high similarity, with a root mean square deviation (RMSD) of 0.63 Å over 976 equivalent C α atoms”

“The two structures exhibit high similarity and the root mean square deviation (RMSD) between them is 0.63 Å”

References:

- 1 Horton, N. & Lewis, M. Calculation of the free energy of association for protein complexes. *Protein Sci.* **1**, 169-181 (1992).
- 2 Vangone, A. & Bonvin, A. M. J. J. Contacts-based prediction of binding affinity in protein–protein complexes. *eLife* **4**, e07454 (2015).
- 3 Kundrotas, P. J. & Alexov, E. Electrostatic Properties of Protein-Protein Complexes. *Biophys. J.* **91**, 1724-1736 (2006).
- 4 Li, W. *et al.* Broad receptor engagement of an emerging global coronavirus may potentiate its diverse cross-species transmissibility. *Proc. Natl. Acad. Sci. U. S. A.* **115**, E5135 (2018).
- 5 Santiago, C. *et al.* Allosteric inhibition of aminopeptidase N functions related to tumor growth and virus infection. *Sci. Rep.* **7**, 46045 (2017).